# Optimal Self-Consistency for Efficient Reasoning with Large Language Models

**Austin Feng** [1]  **Marius Alonso** [2]  **Ambroise Odonnat** [2 3]  **Vasilii Feofanov** [2 4]  **Ievgen Redko** [2]

## Abstract

Self-consistency (SC) is a widely used test-time inference technique for improving performance in chain-of-thought reasoning. It consists of generating multiple responses, or "samples," from a large language model (LLM) and selecting the most frequent answer. This procedure can naturally be viewed as a majority vote or empirical mode estimation. Despite its effectiveness, self-consistency is prohibitively expensive at scale when naively applied to datasets, and it lacks a unified theoretical understanding of sample efficiency and scaling behavior. In this paper, we provide the first comprehensive analysis of SC's scaling behavior and its variants, drawing on mode estimation and voting theory. We derive and empirically validate power law scaling for self-consistency across datasets, and analyze the sample efficiency for fixed-allocation and dynamic-allocation sampling schemes. From these insights, we introduce `Blend-ASC`, a novel variant of self-consistency that dynamically allocates samples to questions during inference, achieving state-of-the-art sample efficiency. Our approach uses $4.8\times$ fewer samples than vanilla SC on average, outperforming both fixed- and dynamic-allocation SC baselines, thereby demonstrating the superiority of our approach in terms of efficiency. In contrast to existing variants, we note that Blend-ASC is hyperparameter-free, supports batching, and can fit any budget of samples, ensuring it can be easily applied to any self-consistency application.

## 1. Introduction

Test-time inference has emerged as a promising direction for improving the performance of large language models (LLMs) on reasoning-intensive tasks (Wei et al., 2022; Snell et al., 2025). These techniques encourage models to "think more" by either exploring diverse reasoning paths (Yao et al., 2023) or producing longer outputs (Muennighoff et al., 2025). Among such approaches, *Self-Consistency* (SC) (Wang et al., 2023), also known as Vote @ $n$, has become widely adopted due to its simplicity and efficiency. For each question, it suffices to sample $n$ chain-of-thought generations and select the most frequent answer. In other words, SC is equivalent to a plurality vote across the sampled outputs, and can be viewed as selecting the empirical mode of the LLM's answer distribution. Beyond improving the accuracy of LLMs, SC was also successfully used for preference optimization (Prasad et al., 2025) and enhancing the reliability of LLMs (Novikova et al., 2025), making it an active research topic as confirmed by numerous recent publications (Huang et al., 2024; Zhang et al., 2024; Cheng et al., 2025; Abdulaal et al., 2025).

Despite the clear ties of SC to mode estimation and voting theory, most attempts to improve or analyze it have relied on ad-hoc statistical methods or semantic approaches (Du et al., 2025; Chen et al., 2023), often overlooking insights from the rich existing literature. The absence of these fundamental approaches leaves SC and its variants without a principled analysis of their sample efficiency, as well as provable guarantees. Yet such an analysis is essential, since SC can be highly inefficient at scale. Under a fixed sampling budget, vanilla SC distributes samples uniformly across questions, regardless of their difficulty. The efficiency could be improved by allocating samples adaptively, focusing more on harder questions (Aggarwal et al., 2023; Li et al., 2024). While such adaptive variants of SC are able to dramatically enhance efficiency, they remain rather underexplored. To address this gap, this work provides a comprehensive analysis of the sample efficiency of SC and its related variants using mode-estimation and voting theory results.

We show that SC follows power-law scaling and identify variants with accelerated and even exponential error decay. Following our analysis, we introduce `Blend-ASC`, a novel adaptive SC algorithm that achieves the best empirical sample efficiency. Motivated by existing mode estimation results, our algorithm matches the initial performance of existing SC variants in the low-sample regime and outperforms all variants at scale. To ensure ease of use for practition-

---

[1]Yale University, New Haven, CT, USA; Work done during an internship at Noah's Ark Lab [2]Noah's Ark Lab, Paris, France [3]Inria, Rennes, France [4]42.com, Amsterdam, Netherlands. Correspondence to: Austin Feng <austin.feng@yale.edu>.

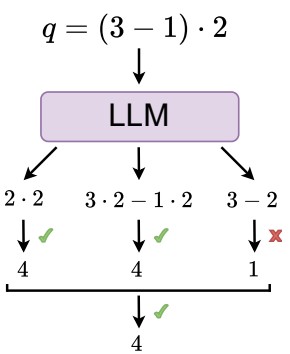 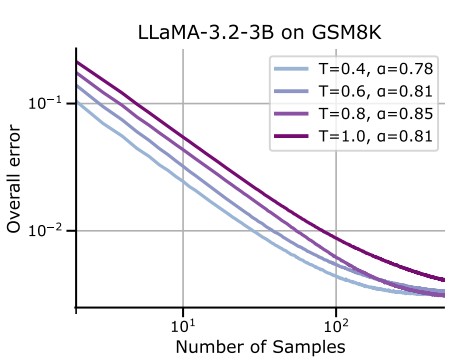 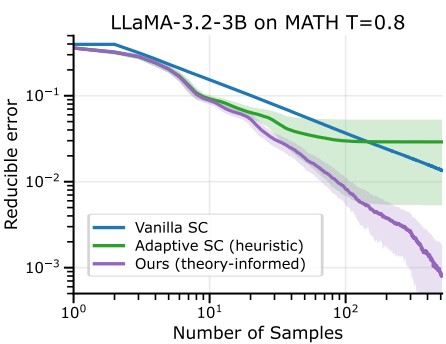

*Figure 1.* (Left) Illustration of the vanilla self-consistency (SC) approach. (Middle) We derive and experimentally validate scaling laws of SC in our work. (Right) `Blend-ASC` converges to the limiting answer the fastest.

ers, we also adapt our method and existing methods to be hyperparameter-free. Ultimately, our algorithm leads to a significant improvement in sample efficiency, requiring $4.8\times$ fewer samples on average than vanilla SC.

Our contributions are as follows:

1. **Theoretical analysis:** We provide a first complete study establishing test-time scaling laws for SC, fixed-allocation SC, and dynamic-allocation SC. Our theoretical results are tighter than previous work on per-question scaling (Huang et al., 2026) and are the first to cover fixed and dynamic allocation SC.

2. **Empirical validation:** We validate our theoretical results using three recent LLMs of varying sizes on three popular reasoning benchmarks, demonstrating their real-world applicability.

3. **Theory-informed SC:** Following our theoretical analysis, we introduce `Blend-ASC`, a SC variant that achieves optimal sample-efficiency for a given budget (Figure 1 (right)), without relying on expensive hyperparameter tuning.

## 2. Self-Consistency as Mode Estimation and Majority Vote

**Setup.** Let $q$ be an input question, fed to an LLM that outputs a chain-of-thought (CoT) yielding a final answer $r$. Let $\mu(\cdot\,|\,q)$ be the distribution of such answers. We say that the LLM is aligned to a question $q$ if the true response $r^*$ is the mode of the LLM distribution, i.e., $r^* = \operatorname{argmax}_r \mu(r\,|\,q)$, and misaligned otherwise. Self-Consistency (SC), illustrated in Figure 1 (left), samples $x$ CoT generations $r_1, \ldots r_x \sim \mu(r\,|\,q)$ and the output is the most frequent answer. In other words, an empirical distribution $\hat{\mu}_x$ is generated based on $x$ sampled answers, and the output of SC is the empirical mode $r_{\mathrm{SC}} = \operatorname{argmax}_r \hat{\mu}_x(r\,|\,q) = \operatorname{argmax}_r \sum_{i=1}^x \mathbb{1}[r_i = r]$. One can notice that if the model is aligned to $q$, SC converges to the

correct response as $x \to \infty$ as $\hat{\mu}_x(\cdot\,|\,q)$ converges to $\mu(\cdot\,|\,q)$. Otherwise, SC converges to an incorrect response, which implies the model is inherently not capable of answering question $q$. Thus, SC sample efficiency is measured by the rate of convergence to the true mode. From a voting theory perspective, we can view the support of $\mu$ as a list of candidates, and each response $r_i$ is a vote from i.i.d voters who select candidates with probability $\mu(\cdot\,|\,q)$.

**Per-question scaling.** To understand the non-asymptotic behavior of self-consistency, we analyze its convergence rate on a single question. Considering an aligned question $q$, we upper bound the expected error of SC defined by

$$\operatorname{err}(x, q) = \mathbb{P}[r_{\mathrm{SC}} \neq r^*]$$
$$= \mathbb{P}\left[\operatorname{argmax}_r \hat{\mu}_x(r\,|\,q) \neq \operatorname{argmax}_r \mu(r\,|\,q)\right].$$

We call $\operatorname{err}(x, q)$ over aligned questions $q$ the *reducible error*. For misaligned questions, the goal is to derive the upper-bound on the accuracy as measured by $1 - \operatorname{err}(x, q)$. This quantity is expected to decrease as it is calculated over questions to which the LLM provides the wrong answer.

Existing voting theory results grow unbounded as the number of candidates, or *unique* responses, increases (Aeeneh et al., 2025; Hu et al., 2024). However, LLMs can vacuously produce infinite unique responses.[1] We extend the error bound in Aeeneh et al. (2025) to handle numerous candidates. We address this by grouping the tail of low-probability responses in $\mu(\cdot\,|\,q)$, deriving a stronger bound by bounding $\mathbb{P}\left[r_{\mathrm{SC}} \neq r^*\right]$ over top $k \ll K$ answers.

**Theorem 1.** *Without loss of generality, we consider the unique responses sorted according to $\mu(\cdot\,|\,q)$ in descending order, denoting their corresponding prob-*

---

[1]For a free-response math question, the set of unique responses could be the set of integers.

[2]We use $x \geq 4$ for GPQA-Diamond for sufficient sample size.

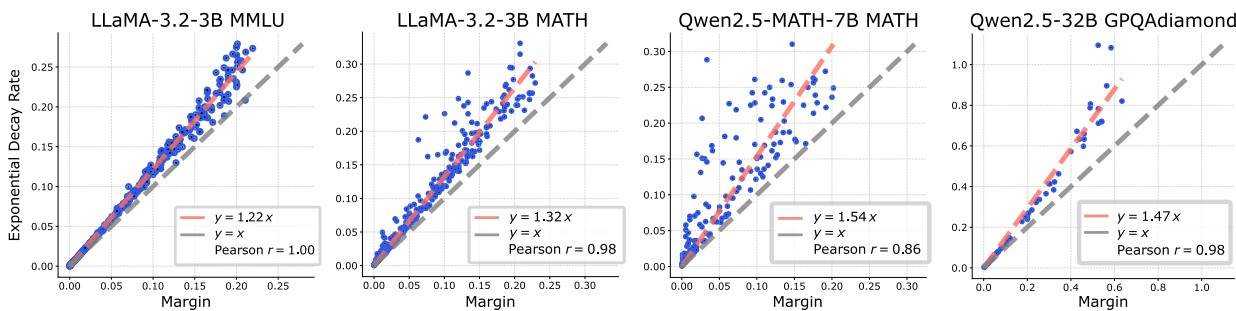

*Figure 2.* **Margin decay.** Margin correlates with decay rate across several model and dataset combinations, where decay is fit for $x \geq 16$ for $\epsilon$ to have negligible impact on the bound.[2]

abilities by $p_1 \geq p_2 \geq p_3 \geq \dots$. Let $\bar{p}_k = \sum_{i \geq k} p_i$ with $k$ that satisfies $\bar{p}_k < p_2$. Then, for the empirical distribution $\hat{\mu}_x$ from $x$ samples, the reducible error satisfies

$$\text{err}(x,q) \leq \exp(-x((\sqrt{p_1} - \sqrt{p_2})^2 + \epsilon))$$

with $\epsilon \to 0$ as $x \to \infty$ with rate $O\left(\frac{\log x}{x}\right)$. For misaligned $q$, the bound holds for $1 - \text{err}(x,q)$.

The proof is deferred to Section C.1. By restricting $k$ to the smallest integer such that $\bar{p}_k < p_2$, we substantially reduce the effective number of candidates. In general, Theorem 1 demonstrates exponential error convergence with rate $m = (\sqrt{p_1} - \sqrt{p_2})^2$, which we refer to as the *margin*. The margin reflects model confidence by quantifying the gap between its most likely answer and the second most likely one. Note that Li et al. (2025) and Huang et al. (2026) also consider the notion of margin, while in a broader range of unsupervised learning tasks, it is common to rely on empirical margin distributions, even when the estimates may be potentially noisy (Feofanov et al., 2024; Xie et al., 2024).

To validate Theorem 1, we consider three recent models, Llama-3.2-3B, Qwen-2.5-Math, and Qwen-2.5-32B, evaluated on MMLU, MATH, and GPQA-Diamond. For each question, we estimate the margin using 100 samples. We calculate the empirical decay rate by fitting an exponential curve to the mode estimation error. Figure 2 shows that all models' performance exhibits a clear correlation between the margin and the error decay rate with an increasing number of voting samples. This behavior is consistent across models and benchmarks as illustrated in Appendix D.

**Comparison with prior work.** Huang et al. (2026) introduces a per-question sample-efficiency bound, stating that $x \geq 2 \log(\frac{1}{\delta})/(p_1 - p_2)^2$ many samples achieve an error of $\delta$. Our result achieves error less than or equal to $\delta$ when $x \geq \log(\frac{1}{\delta})/((\sqrt{p_1} - \sqrt{p_2})^2 + \epsilon)$. We have a tighter result when $2(\sqrt{p_1} - \sqrt{p_2})^2 + \epsilon \geq (p_1 - p_2)^2$. To show this, we note that $\epsilon \to 0$ as $x$ increases, so we consider the simplified bound with $\epsilon = 0$, where it always holds. We then have that

$$\frac{1}{(\sqrt{p_1} - \sqrt{p_2})^2} = \frac{(\sqrt{p_1} + \sqrt{p_2})^2}{(p_1 - p_2)^2} \leq \frac{2\sqrt{p_1}^2 + 2\sqrt{p_2}^2}{(p_1 - p_2)^2} \leq \frac{2}{(p_1 - p_2)^2},$$

which implies our result. This suggests that margin is a more natural measure of confidence in SC compared to the absolute difference $p_1 - p_2$. Besides the per-question bound, we provide a general analysis on the dataset setting to improve SC efficiency, which they do not explore.

**Majority vote assumption.** Our current framework assumes that candidate answers can be aggregated through a majority-vote-style rule. As a result, Blend-ASC and its theoretical guarantees do not directly apply to tasks where aggregation is unnatural or ill-defined, such as open-ended generation.

## 3. Scaling Laws on Dataset Performance

We now broaden our analysis by providing the first test-time scaling law for sample efficiency of SC over benchmarks and datasets, considering a theoretical error model based on the margin $m = (\sqrt{p_1} - \sqrt{p_2})^2$. Such a study is more informative than individual questions considered by prior work, as it provides insights into the empirical behavior of SC in a more common setting where sample efficiency can be meaningfully improved.

**Assumptions.** To study the reducible error over a dataset theoretically, we now consider a dataset $\mathcal{D}$ of infinitely many questions, $(q_i)_{i \in \mathbb{N}}$ with margins $m_i$. Using the theoretical error model $\text{err}(x, q_i) = \exp(-m_i x)$ from Theorem 1, the expected dataset error with $x$ samples for each question is

$$\text{err}(x, \mathcal{D}) = \mathbb{E}_{q_i \sim \text{Unif}(\mathcal{D})}\left[\exp(-m_i x)\right]$$
$$= \int_0^1 e^{-m_i x} p_{\mathcal{D}}(m) dm$$

which is precisely the Laplace Transform $\mathcal{L}\{p_{\mathcal{D}}(m)\}$ where $p_{\mathcal{D}}(m)$ is the probability density function of margin across $\mathcal{D}$. Note that $\mathcal{L}\{p_{\mathcal{D}}(m)\}$ scales as $x^{-1/2}$ if $p(m) \propto m^{-1/2}$ and scales as a power law if $p(m)$ does as well and the exponent is greater than $-1$. We prove that we only need $p_{\mathcal{D}}(m) \propto m^{-1/2}$ near 0 to have $x^{-1/2}$ scaling (see Lemma 5 in Appendix).

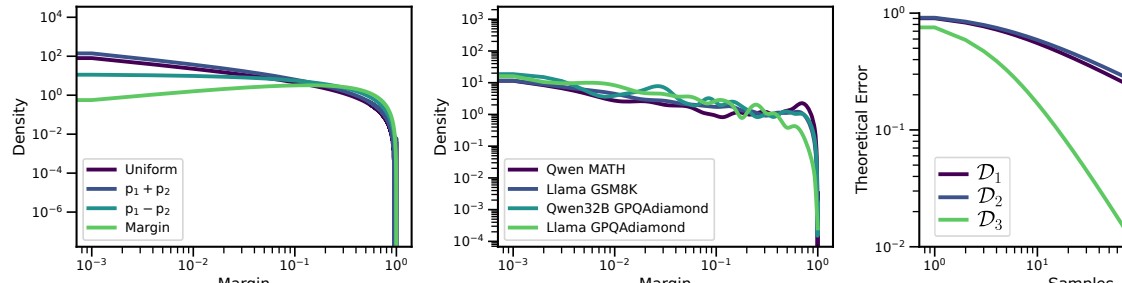

*Figure 3.* **Large dataset sizes induce power-law scaling.** (Left) Margin distribution for $\mathcal{D}_1, \mathcal{D}_2$ and $\mathcal{D}_3$ with $n = 1$. (Middle) Margin distribution by sampling 100 points from each real-world dataset and applying kernel density estimation. (Right) Error scaling for $\mathcal{D}_1, \mathcal{D}_2$ and $\mathcal{D}_3$, with $\mathcal{D}_3$ having the fastest convergence.

Further, we consider the following families of datasets $\mathcal{D}$:

– $\mathcal{D}_1$. Distribution of top two probabilities $(p_1, p_2)$ is uniform across $A = \{(x, y) \mid 0 \leq y \leq x \leq 1, x + y \leq 1\}$, i.e., $g(p_1, p_2) : A \to \mathbb{R}_{\geq 0}$, $g = \mathsf{Unif}(A)$.

– $\mathcal{D}_2$. Distribution of $(p_1, p_2)$ is weighted by $(p_1 + p_2)^n$ for $n > 0$, i.e., $g(p_1, p_2) \propto (p_1 + p_2)^n$. This arbitrarily downweights questions where both $p_1$ and $p_2$ are low, which are questions where the model has low confidence and considers several responses.

– $\mathcal{D}_3$. Distribution $g(p_1, p_2) = (\sqrt{p_1} - \sqrt{p_2})^{2n}$ for $n > 0$. We refer to this case as *adversarial*, since it arbitrarily down-weights low-margin questions, encouraging faster convergence.

To verify that these families are realistic, we sample up to 100 questions from several common models and benchmarks and apply a kernel density estimation to "simulate" a continuous margin distribution. The obtained results, illustrated in Figure 3 (middle), are very close to those observed for datasets $\mathcal{D}_1$ and $\mathcal{D}_2$, as portrayed in Figure 3 (left). This suggests that our theoretical setup is realistic enough to provide insights about the performance of SC in real-world benchmarks.

**Main Result.** Given our margin-dependent theoretical error model, in Proposition 2 we show that for $\mathcal{D}_1$ and $\mathcal{D}_2$, the margin naturally leads to power law scaling, encouraging $p_{\mathcal{D}}(m) \propto m^{-1/2}$ near 0. In contrast, for $\mathcal{D}_3$, the convergence is faster and defined by $p_{\mathcal{D}}(m) \propto m^{-n-1/2}$.

> **Proposition 2.** *For a broad class of datasets* $\{\mathcal{D}_1, \mathcal{D}_2\}$, *the margin distribution satisfies* $\lim_{m \to 0^+} p(m) \propto \frac{1}{\sqrt{m}}$. *For an adversarial dataset* $\mathcal{D}_3$, *we have* $\lim_{m \to 0^+} p(m) \propto m^{-n-1/2}$.

In Figure 3 (right), we show the established theoretical error decay for the three dataset families considered above. We observe power law scaling in both margin and error decay,

consistent with what our theory predicts. As expected, the adversarial dataset $\mathcal{D}_3$ exhibits a faster convergence due to its favorable reweighting of low-confidence questions.

**Empirical results.** Finally, we provide a more nuanced illustration of the observed scaling law on real-world datasets in Figure 4 (additional graphs are in Section E). We plot the true error rate across multiple-choice and free-response benchmarks with Llama-3.2-3B on GSM8K and Qwen-32B on GPQA-Diamond across temperatures ranging from 0.4 to 1. In each plot and for each temperature, we report the slope of the scaling law as $\alpha$. We further examine the performance on the subsets of aligned questions (left) and misaligned questions (right), where the model, on average, provides a correct and incorrect answer, respectively.

Our first observation is that for correct answers (left), we have extremely strong power-law scaling, with weaker power-law scaling for incorrect answers (middle). For full datasets (right), we combine both contributions from both correct and incorrect answers. The full dataset plots show a consistent power-law scaling for free-response questions (top row), and a less monotonic scaling for multiple-choice questions (bottom row).

Our contributions for dataset-level performance are well-aligned with the empirical observations made by Schaeffer et al. (2025), who showed that the test-time inference performance scaling in the case of multiple-choice tasks is difficult to predict. We note that this is also the case when studying this problem theoretically. Obtaining a tight bound for the datasets with dominating incorrect answers requires an infeasible combinatorial analysis that depends on the number of possible rankings between the most likely answer from the LLM's perspective and the true answer, as well as the probabilities of each answer in between. We leave this analysis as an exciting open problem for future work.

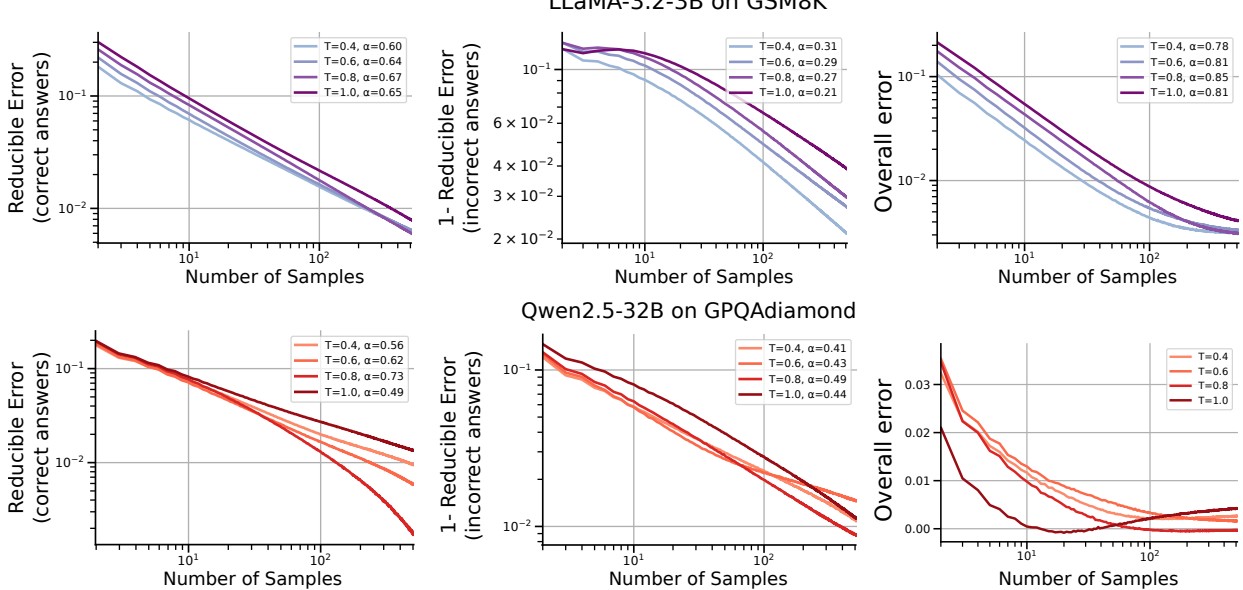

*Figure 4.* **Scaling behavior of Self-Consistency:** correct answers (left), wrong answers (middle), and full (right) datasets for free-response (top) and multiple-choice (bottom) benchmarks.

## 4. Scaling Laws for Adaptive Self-Consistency

A major drawback of SC when used on datasets is sample efficiency. Some questions only need a few samples, while others require hundreds, yet we use the same number of samples per question for the whole dataset. This motivates *adaptive SC*, which allocates samples per question given some budget of total samples. We consider two settings: *fixed allocation*, where samples are allocated a priori, and *dynamic allocation*, where samples are allocated during inference. Fixed methods can be used when there is some information about questions. Dynamic methods rely on consistency in the sample empirical distribution to inform mode stability.

### 4.1. Fixed Allocation

In scenarios where we possess a priori information that influences model performance, such as question difficulty or subject, we can adjust the number of samples per question accordingly. As an example, Wang et al. (2025b) allocates just one sample for "easy" questions. To quantify the sample efficiency of fixed allocation, we consider the optimal setting where we have full information on the question, or oracle access to $\mu$.

We assume that $\mathrm{err}(x, q_i) = \exp(-m_i x)$ and let $x_{q_i}$ be the number of samples allocated to question $q_i$ and $\bar{x}$ be the average samples per question. Then we have $\mathrm{err}(\bar{x}, \mathcal{D}) = \mathbb{E}_{q_i \sim \mathsf{Unif}(\mathcal{D})}[\exp(-m_i x_{q_i})]$. Since error only depends on margin, oracle access to $\mu$ means that we can access the margin $m_i$ for any $q_i$. So based on $m_i$, we should

optimally choose $x_{q_i}$. We see that all questions with the same margin should have the same number of samples, so we can let $x_{q_i} = x_{m_i}$ be a function of margin. Suppose now that we have two questions $q_i$ and $q_j$ with the same margin $m_i = m_j$. The total error is $\exp(-m_i x_{q_i}) + \exp(-m_i x_{q_j}) \geq 2\sqrt{\exp(-m_i x_{q_i})\exp(-m_i x_{q_j})} = 2\exp(-m_i(x_{q_i} + x_{q_j})/2)$. So it is optimal to distribute samples equally among such tied questions. Then, we define $x_m \in M \subset \{f \mid f : (0,1] \to \mathbb{N}\}$ where $M$ is the set of functions from $(0,1]$ to $\mathbb{N}$ such that $\bar{x} = \int_0^1 x_m p(m)dm$ for some average number of samples $\bar{x}$. We can express the error as

$$\mathrm{err}(x_m, \mathcal{D}) = \min_{x_m \in M} \mathbb{E}_{q_i \sim \mathsf{Unif}(\mathcal{D})}[\exp(-m_i x_{m_i})]$$

$$= \min_{x_m \in M} \int_0^1 \exp(-m x_m)p(m)dm,$$

which becomes a constrained convex optimization problem. For a tractable, closed-form solution, we weaken our assumption to have $x_i \geq 0$ (so $x_i$ need not be an integer) and solve this problem by means of the following proposition.

**Proposition 3.** *Under the above assumptions and with $p(m) \propto m^{-r}$ for $r \in (0,1)$, the optimal sample allocation is*

$$x_m = \begin{cases} m^{-1}(\log m - \log \lambda) & \text{if } m \geq \lambda \\ 0 & \text{if } m < \lambda \end{cases}$$

*where as $\bar{x} \to \infty$, $\lambda \sim \bar{x}^{-\frac{1}{r}}$. This gives us an error convergence rate of approximately $\bar{x}^{-\frac{1-r}{r}}$ which becomes $\bar{x}^{-1}$ in the special case of $r = \frac{1}{2}$.*

Theorem 3 suggests that for a sufficiently small margin ($m < \lambda$), it is no longer efficient to allocate any samples. This is because the marginal improvement of a single sample is less than adding a sample to a question with a higher margin. We illustrate our obtained results and the above-mentioned intuition of optimal fixed allocation in Figure 5. We use the error model $e^{-m_i x_{m_i}}$ with margins extracted from running Llama-3.2-3B on MATH.

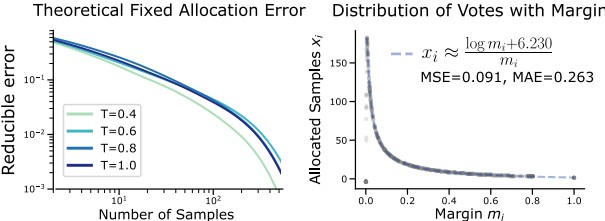

*Figure 5.* **Optimal allocation.** (Left) Fixed-Allocation SC scaling. (Right) The number of allocated samples closely follows the theoretical distribution depending on the margin.

The obtained scaling law across different temperatures confirms the accuracy of our theoretical result in the large sample regime. Finally, Figure 5 (right) shows the impact of the margin on fixed allocation: low-margin questions where the LLM is uncertain get more samples (margin is close to 0 on the x-axis), while the number of samples for high-margin (rightmost part of the x-axis) is fewer.

### 4.2. Dynamic Allocation

Instead of assigning a fixed number of samples a priori for each question, *dynamic allocation* adaptively samples during inference until a stopping criterion $\mathcal{S}_\delta$ indicates that the self-consistency output $r_{\text{SC}}$ achieves high confidence. Following Shah et al. (2020), these criteria require at least $x(\mathcal{S}_\delta, q_i) = \Omega(\ln 1/\delta)$ samples for question $q_i$, where $\delta \in (0, 1)$ denotes the target expected error.

The theoretical bridge between Self-Consistency and mode estimation allows us to build upon a rich literature on martingale confidence sequences to find an optimal stopping criterion $\mathcal{S}_\delta$. In particular, we can use the PPR-1v1 stopping criterion (Anand Jain et al., 2022) that has a theoretically optimal *exponential* decay in error for predicting the mode. At each iteration, we allocate a sample to the question with the lowest confidence, evaluated by $(K-1)\text{Beta}(x, n_1+1, n_2+1)$ where $K$ refers to the number of possible *unique* answers, and $n_1$ and $n_2$ are counts for the two most frequent answers. For target error $\delta$, the stopping criterion is defined as $\text{Beta}(\frac{1}{2}, n_1 + 1, n_2 + 1) \leq \frac{\delta}{K-1}$.

In Corollary 4, we provide a scaling law for adap-

tive SC with a theoretically optimal allocation across datasets.

**Corollary 4.** *Given a dataset $\mathcal{D}$, a target error $\delta \in (0, 1)$ and stopping condition PPR-1v1 denoted as $\mathcal{S}_\delta$, we have that*

$$\lim_{\delta \to 0^+} \mathbb{E}_{q_j \sim \text{Unif}(\mathcal{D})} \left[ \frac{x(\mathcal{S}_\delta^P, q_j)}{\Omega(\ln 1/\delta)} \right] = 1.$$

Corollary 4 establishes the theoretically optimal decay rate for dynamic allocation and shows that the PPR-1v1 algorithm achieves this rate following the lower bound from (Shah et al., 2020). The groundedness of PPR-1v1 in theory makes it principally different from the other heuristic-based adaptive methods (Li et al., 2024; Aggarwal et al., 2023) for which no theoretical guarantees are available.

### 4.3. Putting Theory to Work: `Blend-ASC`

Despite the theoretical guarantees for the PPR-1v1 method, we have observed empirically (see Figure 7) that in the low-sample regime, it tends to be pessimistic, and simpler policies like ASC (Aggarwal et al., 2023) remain very efficient. This motivates us to introduce `Blend-ASC` that combines the best of the two worlds. For each question $Q$, we recover the confidence score from ASC and PPR-1v1 with associated rankings $\text{ASC}(Q)$ and $\text{PPR}(Q)$, respectively. Then, at step $t$ out of $T$ samples, we generate a response for the question that minimizes the linearly-scaled ranking $(1 - \frac{t}{T})\text{ASC}(Q) + \frac{t}{T}\text{PPR}(Q)$. This is summarized in Figure 6. We observe that PPR-1v1 overly concentrates samples on a few questions, so we exclude questions with over 16 times the number of samples than the average question.

**Hyperparameter-free and fixed budget.** Prior methods require tuning hyperparameters, often related to the threshold used for the stopping criterion, which may reduce the computational gains obtained when using them. For instance, the original ASC considers a Beta prior on the distribution of $\frac{p_1}{p_1+p_2}$, and retrieves its posterior distribution given observed counts $n_1$ and $n_2$. Then the stopping condition enforces that the probability of $p_1 < p_2$ is lower than some threshold $\tau$ which needs to be fixed by the user. In contrast, we propose to jointly allocate samples across all questions and then assign samples to questions with low confidence as explained in Figure 6. As a direct consequence of our method being hyperparameter-free, it allows the user to strictly enforce the provided budget $T$. For both ASC and its close variation ESC (Li et al., 2024), the average number of samples is determined by hyperparameters, and even with the same set of parameters, the number of samples on each instance is stochastic.

**Repeat until the budget ($t < T$) is reached**

Low-sample optimal

Asymptotically optimal

1. Compute Blend-ASC$(Q) = \left(1 - \dfrac{t}{T}\right)$ ASC$(Q) + \dfrac{t}{T}$ PPR$(Q)$

Dominates for small $t$

Dominates for large $t$

2. Choose new hardest question $Q = \text{argmin}_Q \texttt{Blend-ASC}(Q)$

3. Call LLM to generate a new CoT for $Q$

*Figure 6.* Overview of Blend-ASC.

*Table 1.* **Sample efficiency of adaptive methods.** We compare how many samples are required to achieve a reducible error lower than SC using 64 and 128 samples. ASC doesn't reach the target SC accuracy for Llama on GSM8K, so we let the samples be 128.

| SC@n | Algorithm | GSM8K | | MATH | | GPQA-Diamond | | Average Improvement |
|---|---|---|---|---|---|---|---|---|
| | | Llama-3B | Qwen-Math | Llama-3B | Qwen-Math | Llama-3B | Qwen-32B | |
| 64 | Fixed-Allocation | 15 | 9 | 18 | 10 | 27 | 15 | 4.09× |
| | Adaptive SC | 13 | 6 | 16 | 7 | 33 | 13 | 4.36× |
| | **Blend-ASC (Ours)** | 11 | 6 | 14 | 7 | 26 | 8 | **5.33×** |
| 128 | Fixed-Allocation | 34 | 14 | 45 | 19 | 77 | 37 | 3.40× |
| | Adaptive SC | 128 | 11 | 77 | 13 | 102 | 29 | 2.13× |
| | **Blend-ASC (Ours)** | 22 | 9 | 31 | 12 | 80 | 25 | **4.29×** |

**Computational overhead.** We note that using the stopping condition proposed by our approach represents a negligible computational overhead. Indeed, it takes 1.8 seconds to run Blend-ASC on GPQA-Diamond with a budget of 128 samples. From OpenRouter, the throughput of GPT-5 is 37.12 tokens per second. A single chain-of-thought can easily be over $37.12 \cdot 1.8 < 70$ tokens long, exceeding the total time of Blend-ASC, and the GPQA-Diamond task requires 25000 such chain-of-thought responses.

**Related work.** We note that despite extensive adaptations to SC (Wang et al., 2025a; Taubenfeld et al., 2025), there are few papers analyzing SC behavior and sample efficiency both empirically and theoretically. Chen et al. (2024) showed that there is often no monotonic increase in SC performance with samples on multiple-choice benchmarks. Ruan et al. (2024) uses "observational" scaling laws which fit curves across several LLMs to predict SC behavior, but observe a weak scaling trend with FLOPs. From a theoretical point of view, Hu et al. (2024) introduced a per-question bound on error, but the bound scales with the number of unique reasoning steps, which can be vacuously large. Huang et al. (2026) provides a per-question bound similar to Theorem 1 with detailed comparisons in the dedicated section, but they are limited to the per-question setting and do not explore improving efficiency. Finally, we note that several other extensions to SC were recently proposed in the research community that use the LLM to reflect on the

initial responses in order to rectify them if needed. Some representative examples of these include mirror-consistency (Huang et al., 2024), self-check (Miao et al., 2024), and self-contrast (Zhang et al., 2024). Such approaches, often based on enhancing the reasoning capability of the model, are sequential in nature, and they present a complementary axis of test-time scaling to parallel scaling explored in this work. Indeed, as mentioned by (Snell et al., 2025), on hard benchmarks, the best performance is achieved with some ideal ratio of sequential to parallel computing.

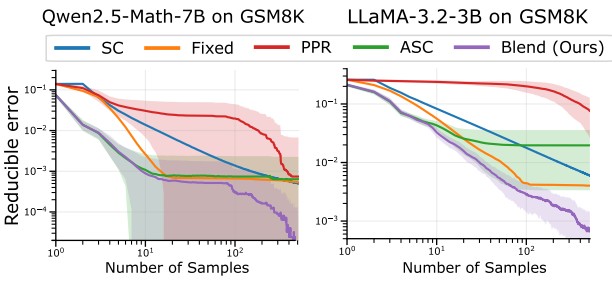

*Figure 7.* **Performance comparison.** Blend-ASC consistently outperforms all methods in mode-estimation across models and datasets, achieving the lowest sample efficiency for target error.

## 5. Experiments

**Benchmarks and models.** We evaluate our findings using a variety of models, including Llama-3.2- (Grattafiori et al., 2024), Qwen2.5-MATH-7B (Yang et al., 2024), and

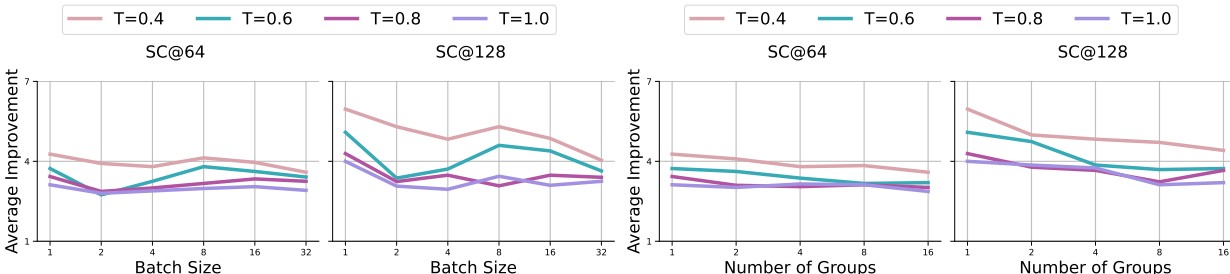

*Figure 8.* **Extended analysis.** Left: batched prompts are used where a batch of questions of size $b$ is selected in step 2 before running the inference in step 3 of Figure 6. Right: batched queries are used where `Blend-ASC` is applied on batches of queries, each seen as an independent dataset. For each pair of model and dataset from Table 1, we count the number of samples that `Blend-ASC` uses to achieve the same reducible error as SC at 64 or 128 samples. The performance reported is the average improvement in terms of sample efficiency over the 6 pairs of model-dataset. For both extensions, we observe stable performance for each temperature $T \in \{0.4, 0.6, 0.8, 1.0\}$.

Qwen2.5-32B, and temperature settings from $0.4$ to $1.0$. We evaluate on both free response and multiple choice datasets, including GSM8K (Cobbe et al., 2021), MATH (Hendrycks et al., 2021), MMLU (Hendrycks et al., 2020), and GPQA-Diamond (Rein et al., 2023). Due to the high computational cost of scaling test-time inference, directly running inference can be prohibitively expensive in time and compute. So, we sample 100 generations per question from an LLM to form the "true" LLM distribution. SC is performed by sampling from the corresponding multinomial distribution. Following Huang et al. (2026), we focus on aligned questions where the LLM is capable of attributing a high probability to the right answer. Our goal then is to improve the sample complexity of SC over such questions to avoid unnecessary LLM calls.

**Baselines.** We use hyperparameter-free ASC and PPR-1v1 as dynamic allocation baselines, each run 100 times per model and benchmark pair. Fixed allocation SC is intractable as our problem is a non-convex integer programming problem, so we use a modified method. We leave implementation details in Section B. To assess the sample efficiency of SC variants, we calculate the error of SC at 64 and 128 samples, and then identify the least number of average samples required to match the error for each variant. If a method does not reach the desired error, we set the samples used to 64 or 128.

**Results.** In Table 1, we observe that `Blend-ASC` consistently achieves optimal performance (see Appendix F for more plots across models, benchmarks and temperatures). SC has reliable performance improvements, but performs worse on the low-sample regime. Fixed-Allocation SC performs strongly, reducing samples by 3.75 times compared to SC. But it is only competitive with the unreasonable assumption of complete oracle access to $\mu$, which is never available, and underperforms ASC and `Blend-ASC`. This suggests that allocating samples before inference using prior information is never optimal. With our parameter-free and fixed budget modifications, dynamic allocation is also easier to use than allocating samples beforehand.

For dynamic allocation, PPR-1v1 is the worst method in the low-sample regime but can beat ASC in the large-sample regime, as its theoretical exponential performance suggests. ASC performs strongly in the low-sample regime but often gets stuck at large samples, highlighting its weakness in having no theoretical convergence guarantees. Finally, `Blend-ASC` matches ASC in the low-sample regime but dominates as we scale up samples, reducing the number of prompts by $4.8$ times compared to SC. These performance gains highlight the usefulness of the provided theoretical analysis in practice. Finally, we note that the exponential error decay suggests that we can push the number of drawn samples even further. This is highlighted by the fact that `Blend-ASC` performance is not plateauing even for as many as $10^3$ samples (see Appendix F for visualizations). This can be particularly suited for public model releases where even the slightest performance gain at the expense of a drastically larger compute budget is acceptable.

**Extended analysis.** Finally, we explore extensions of `Blend-ASC` for realistic deployment scenarios. In particular, we study batched aggregation of queries to improve latency and how to adapt to the online nature of queries in many real-world applications.

1. **Batched prompts.** Batching is critical to real-world throughput as it generates LLM responses in parallel. To support this, we simply choose the batch size $b$ questions that minimize `Blend-ASC`$(Q)$ at each iteration at Step 2 and run batch inference at Step 3 of our algorithm. From Figure 8 (left), we observe steady results across batch sizes, which suggests that `Blend-ASC` remains competitive even with batched generations.

2. **Batched queries.** Many situations involve periodically servicing a stream of latency-sensitive queries instead of a fixed dataset. We test a simple modification where we aggregate streamed queries until we reach a certain batch size, and process them as a group using `Blend-ASC`, each with the same budget. To satisfy latency constraints, this baseline must perform well with frequent servicing,

which is the case as can be seen in Figure 8 (right). We progressively increase the number of groups, thereby decreasing the batch size, and observe steady performance.

## 6. Conclusion and Future Work

In this work, we introduce a comprehensive framework based on mode estimation and voting theory, leading to theoretical convergence guarantees and scaling laws. We first analyze SC performance on individual questions and identify power-law scaling behavior across full datasets. With this foundation, we derived improved theoretical sample efficiency results for variants on synthetic datasets, which are validated by empirical results. Finally, we introduced `Blend-ASC`, a novel parameter-free SC variant that combines an asymptotically optimally SC variant with Adaptive SC. Experiments show that `Blend-ASC` consistently outperforms all previous methods.

A natural direction for future work is to extend `Blend-ASC` beyond majority-vote-style aggregation. We believe the main challenge is to define suitable confidence measures for alternative aggregation strategies. For instance, one could aggregate open-ended outputs using embedding space clustering and define confidence through density or cohesion-based measures over clusters (Jain et al., 2024). We can also leverage the mean and median estimation literature to analyze variants that predict continuous values, such as with time-series prediction (Liu et al., 2025).

Some practical components of our approach, such as global, budget-aware allocation, may transfer more broadly across Self-Consistency variants. Establishing analogous theoretical scaling laws in these settings would require assumptions on per-question correctness scaling behavior. For example, theoretical results on weighted majority voting can inspire analysis for variants that use a verifier or LLM to generate scores and then perform majority vote, such as Best-of-N-Weighted or Self-Calibration (Snell et al., 2025; Huang et al., 2025).

Finally, self-consistency can provide a useful signal for preference optimization as shown in Prasad et al. (2025). Our work provides a solid theoretical foundation for making this application of SC as efficient as possible.

## Acknowledgements

The authors would like to thank Simeng Han and Tom McCoy for insightful discussions on Self-Consistency. This work was made possible thanks to open-source software, including Language Model Eval Harness (Gao et al., 2024), PyTorch (Paszke et al., 2019), and vLLM (Kwon et al., 2023).

## Impact Statement

This paper presents work whose goal is to advance the field of Machine Learning. There are many potential societal consequences of our work, none of which we feel must be specifically highlighted here.

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

# Appendix

## Table of Contents

## A. Adaptive SC and Early-Stopping SC

Adaptive Self-Consistency (ASC) specifically looks at the counts of the two most frequent classes, $n_1$ and $n_2$, sampled with probabilities $p_1$ and $p_2$. It considers a Beta prior on the distribution of $\frac{p_1}{p_1+p_2}$, and retrieves its posterior distribution given observed counts $n_1$ and $n_2$. Then the stopping condition activates when the probability of $p_1 < p_2$ is lower than some fixed threshold $\tau$:

$$\mathbb{P}\left[p_1 < p_2\right] = \int_0^{1/2} \text{Beta}\left(x, n_1 + 1, n_2 + 1\right) dx < \tau$$

or we reach some maximum number of samples. We then output the answer corresponding to $n_1$.

Early-Stopping Self-Consistency (ESC) repeatedly samples windows of size $w$ and continues until we reach a maximum number of samples or a window where all samples have the same answer, in which ESC returns that answer. From a mode-estimation perspective, ESC is not theoretically optimal, as it should strictly return the empirical mode regardless of the last window's unique answer (though they often coincide).

Finally, they do not admit a straightforward extension to a setting with a fixed budget of samples, making them unreliable and difficult to use. The average number of samples is determined by hyperparameters, and even with the same set of parameters, the number of samples on each instance is stochastic.

## B. Implementation details

**Fixed allocation.** In practice, the optimal fixed allocation for a true dataset is infeasible as the average error for a question $q$ given $x$ samples is often non-convex in $x$. Since $x$ must be integers, we are thus solving a non-convex integer programming over potentially thousands of variables (sample allocation for each question). However, we observe in Section 2 that the average error for a question highly correlates with a convex and strictly decreasing exponential upper bound.

We can closely approximate Fixed Allocation SC as a convex integer programming problem by allocating samples according to a convex and monotonic approximation, where we apply local smoothing approaches and then fit an exponential approximation for high sample questions. Under these assumptions, for any question $q$, each additional sample yields strictly positive but diminishing improvements in error. Thus, we can greedily allocate samples at each iteration to the question with the largest marginal improvement.

**Dynamic allocation.** To develop hyperparameter-free methods, we create an array for confidences, which is measured for each question based on the consistency of the current samples for that question. Each confidence is initialized to $-\infty$ (or a small initial value). We have another array that stores the sampled responses for each question. Then, at each iteration, we choose the question with the lowest confidence to sample via a heap. After updating our counts array, we update our confidence as follows: for ASC our confidence is $\int_0^{1/2} \text{Beta}(x, n_1 + 1, n_2 + 1) dx$ and for PPR-1v1 our confidence is $(K - 1)\text{Beta}(x, n_1 + 1, n_2 + 1)$.[3] For PPR-1v1 specifically, as we don't have access to $K$ a priori, we modify $K$ to be $\min(2, \hat{K})$ where $\hat{K}$ is the number of unique answers seen thus far while sampling.

---

[3]We modify the confidence in PPR-1v1 for when $K = 1$ and when we only have $n_1 + n_2 = 1$ to avoid degenerate confidences.

# C. Proofs

## C.1. Proof of Theorem 1

We assume the distribution of answers to the question $q$, $\mu(\cdot \mid q)$, has a finite support of $n$ items. We also denote $r_{\max} = \arg\max_r \mu(\cdot \mid q)$ (assuming that $\arg\max$ outputs a single value). Then, following Theorem 3 of Aeeneh et al. (2025), for any $q$, we have

$$\mathbb{P}\left[r_{\mathrm{SC}} \neq r_{\max}\right] \leq \exp\{-x((\sqrt{p_1} - \sqrt{p_2})^2 + \epsilon))\} = (K-1) \cdot \exp\{-x((\sqrt{p_1} - \sqrt{p_2})^2 + \hat{\epsilon}))\}$$

where $\hat{\epsilon}$ has no dependence on $K$, and ties are broken randomly (as assumed in SC). The number of unique answers $K$ can be arbitrarily large (such as the space of integers), which weakens the inequality. Nevertheless, in practice, nearly all probability mass concentrates on a few answers, so we prove that we can bound $\mathbb{P}\left[r_{\mathrm{SC}} \neq r_{\max}\right]$ by truncating to only the top $k \ll K$ answers.

Without loss of generality, let the support of $\mu(\cdot \mid q)$ be $A = \{a_1, \ldots, a_K\}$ with $p_i = \mu(a_i \mid q)$ such that $p_1 \geq \cdots \geq p_K$ and $a_1 = r_{\max}$. Suppose $x$ answers are sampled $r_1, r_2, \ldots, r_x \sim \mu(\cdot \mid q)$ and let the count vector be $(n_1, \ldots, n_K)$ where $n_i = \sum_{j=1}^x \mathbb{1}[r_j = a_i]$. Define the tail bucket distribution $\tilde{\mu}$ of the original $\mu$ by aggregating all answers $a_i$ for $i \geq k$ into $\tilde{a}_k$:

$$\begin{cases} \tilde{\mu}(a_i \mid q) = \mu(a_i \mid q) & \text{if } i < k, \\ \tilde{\mu}(a_i \mid q) = \sum_{j=k}^K \mu(a_j \mid q) & \text{if } i = k. \end{cases}$$

Finally, by $\tilde{r}_{\mathrm{SC}}$ we denote the Self-Consistency answer sampled from $\tilde{\mu}(\cdot \mid q)$.

We prove that $\mathbb{P}\left[r_{\mathrm{SC}} \neq r_{\max}\right] \leq \mathbb{P}\left[\tilde{r}_{\mathrm{SC}} \neq r_{\max}\right]$ by considering all possible count vectors. We can create an injection from the set of count vectors for $\mu$ into the set of count vectors for $\tilde{\mu}$ defined as $(n_1, \ldots, n_K) \to (n_1, \ldots, n_{k-1}, \sum_{j=k}^K n_j)$. This creates a new voting instance where we only have $k$ items and the probability of a count vector in the image is the sum of its pre-image count vector probabilities. It suffices to show that for every count vector from the $\mu$ voting instance is mapped to a count vector in $\tilde{\mu}$ with the same or a higher probability of error.

First, all count vectors where $r_{\mathrm{SC}} = r_{\max}$ is always true ($n_1$ is strictly the largest count) trivially map to count vectors with the same or higher probability of error. [4] Consider all count vectors from $\mu$ where we could have $r_{\mathrm{SC}} \neq r_{\max}$ (under tiebreaks). Then there exists some $i \in \{2, \ldots, K\}$ such that $n_i = \max_j n_j \geq n_1$, where $i$ is the first possible index. If $i < k$, then in the injected vector for $\tilde{\mu}$, $n_i \geq n_1$. The probability of error can only increase between the pre-image and image. If pooling leads to $n_1 > n_k$, the probability of error stays the same (either probability 1 if $n_i > n_1$ or the same tiebreak probability).[5] If $n_k = n_1 \leq n_i$, the probability stays the same at 1 if $n_1 < n_i$ and decreases or stays the same if $n_1 = n_i$ because the size of our tiebreaker increases by 1 or stays the same.[6] If $n_1 < n_k \leq n_i$, the probability of error remains at 1.

But if $i \geq k$, then for $\tilde{\mu}$, we have the final element is $\sum_{j=k}^K n_j \geq n_i \geq n_1$. If we have a strict inequality, then $n_i > n_1$ and both $r_{\mathrm{SC}} \neq r_{\max}$ and $\tilde{r}_{\mathrm{SC}} \neq r_{\max}$. The probability of error remains the same at 1. Consider when we have $n_i = n_1$ and $i \geq k$. We show that in each case, the probability of failure is the same or increases when we inject from $(n_1, \ldots, n_K) \to (n_1, \ldots n_{k-1}, \sum_{j=k}^K n_j)$. Either $\sum_{j=k}^K n_j = n_i$ or $\sum_{j=k}^K n_j > n_i$. In the first case, $|\arg\max_j n_j|$ stays the same (as it must be that everything else in the pool was 0), and in the latter case, we have complete failure for $\tilde{\mu}$. So we see that for all count vectors where we could have $r_{\mathrm{SC}} \neq r_{\max}$, the probability of failure stays the same or increases for $\tilde{\mu}$, so $\mathbb{P}\left[r_{\mathrm{SC}} \neq r_{\max}\right] \leq \mathbb{P}\left[\tilde{r}_{\mathrm{SC}} \neq r_{\max}\right]$.

Then we can bound $\mathbb{P}\left[\tilde{r}_{\mathrm{SC}} \neq r_{\max}\right]$ using the bound from Aeeneh et al. (2025), proving our result.

When our distribution is aligned, $r_{\max}$ is the correct answer, so $\mathrm{err}(x, q) = \mathbb{P}\left[r_{\mathrm{SC}} \neq r_{\max}\right]$ and we upper bound this error. When our distribution is misaligned, then to bound $1 - \mathrm{err}(x, q)$, we notice that whenever SC predicts correctly, $r_{\mathrm{SC}} \neq r_{\max}$

---

[4]For count vectors where $n_1$ is strictly the largest, this case is trivially true as the probability of error is 0, and thus can never decrease.

[5]For tiebreaks, the probability is split equally among items involved. The tiebreak probability is the same as the elements involved all have counts of $n_1$ and thus are in the first $k-1$ entries of the count vector.

[6]If $n_k = 0$ before aggregation (so everything from $k+1$ to $K$ is 0), the tiebreaker size is the same. Otherwise, we added one more item, the pooled $n_k$, to it.

as $r_{\max}$ is an incorrect answer. Then $1 - \text{err}(x, q) \le \mathbb{P}[r_{SC} \ne r_{\max}]$.

## C.2. Proof of Proposition 2

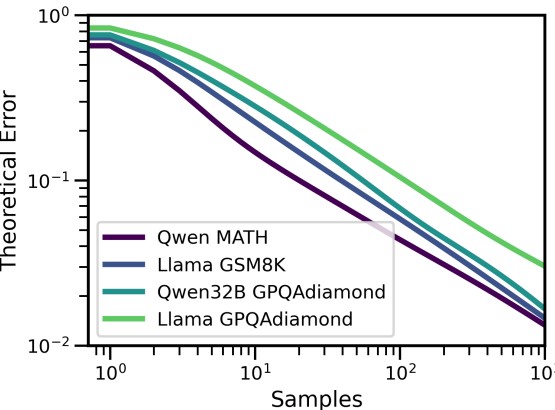

*Figure 9.* From Figure 3, we observe that synthetic dataset $\mathcal{D}_1, \mathcal{D}_2, \mathcal{D}_3$ have power-law scaling under our theoretical error model. Here, we show that our kernel-smoothed distribution has approximately $x^{-1/2}$ error scaling.

We demonstrate that many synthetic datasets exhibit $x^{-1/2}$ scaling under self-consistency by first demonstrating that we only need $p_{\mathcal{D}}(m) \propto m^{-1/2}$ near 0 to have $x^{-1/2}$ scaling. Then, since our error model is only dependent on margin, $m = (\sqrt{p_1} - \sqrt{p_2})^2$, we show that margin naturally encourages $p_{\mathcal{D}}(m) \propto m^{-1/2}$ near 0 by considering various synthetic datasets where the proportion of questions with top two probabilities $(p_1, p_2)$ is uniform across $A = \{(x, y) \mid 0 \le y \le x \le 1, x + y \le 1\}$ or weighted by $(p_1 + p_2)^n$. We demonstrate that even adversarially designed datasets that minimize margin also exhibit power law scaling.

The first lemma demonstrates that we only require $p(m)$ to have $m^{-1/2}$ behavior when $m \in (0, b]$ for small $b$ to get a power law lower bound on $\text{err}(x, \mathcal{D})$.

**Lemma 5.** *Consider the function class*

$$F = \left\{ f(x) \in \mathcal{P}((0, 1]) \mid f(x) = \begin{cases} ax^{-1/2} & \text{if } x \le b \\ h(x) & \text{otherwise} \end{cases}, a > 0, \forall x\, h(x) \ge 0 \right\}$$

*For any $f \in F$, $\mathcal{L}\{f\} = \frac{a}{\sqrt{x}}\gamma(\frac{1}{2}, bx) + O(e^{-bx})$ where $\gamma(\frac{1}{2}, bx) = \sqrt{\pi}(2\Phi(\sqrt{2bx}) - 1)$ is the lower incomplete gamma function and $\Phi$ is the Gaussian cumulative density function and $\Phi(x)$ converges rapidly to 1 with rate $e^{-x^2/2}$.*

*Proof.* The Laplace Transform is

$$\mathcal{L}\{f\} = \int_0^b e^{-tx} \cdot at^{-1/2}dt + \int_b^1 e^{-tx}h(t)dt$$

$\int_b^1 e^{-tx}h(t)dt$ is $O(e^{-bx})$ as the integral of $h$ must be bounded for $f$ to be a distribution. Then we have

$$\mathcal{L}\{f\} = a\int_0^b e^{-tx}t^{-1/2}dt + O(e^{-bx})$$

$$= a\int_0^{bx} e^{-u}\left(\frac{u}{x}\right)^{-1/2}\frac{du}{x} + O(e^{-bx})$$

$$= \frac{a}{\sqrt{x}}\int_0^{bx} e^{-u}u^{-1/2}du + O(e^{-bx}) = \frac{a}{\sqrt{x}}\gamma(\frac{1}{2}, bx) + O(e^{-bx}) \ge \frac{a}{\sqrt{x}}\gamma(\frac{1}{2}, bx)$$

$\square$

We can expand $f$ to be any function lower bounded by $ax^{-1/2}$ when $x \leq b$, and we maintain the same $\frac{a}{\sqrt{x}}\gamma(\frac{1}{2}, bx)$ lower bound, and then we next demonstrate that several synthetic datasets $\mathcal{D}$ have $p(m) \in f$.

**Lemma 6.** *Let the distribution of $(p_1, p_2)$ across $\mathcal{D}$ be $g(p_1, p_2) : A \to \mathbb{R}_{\geq 0}$ where $A = \{(x, y) \mid 0 \leq y \leq x \leq 1, x + y \leq 1\}$. If $g = \mathsf{Unif}(A)$, then $p(m) = \frac{1}{3\sqrt{m}}\sqrt{2-m}^3 - \sqrt{m}\sqrt{2-m} + \frac{2m}{3}$, which implies that $\lim_{m \to 0^+} p(m) \propto \frac{1}{\sqrt{m}}$.*

*Proof.* We consider the cumulative distribution function of $m$, $F(m)$. Let $A_m = \{(x, y) \in A \mid (\sqrt{x} - \sqrt{y})^2 \leq m\}$.

$$F(m) = \iint_{(x,y) \in A_m} g(x, y) dx dy = 4 \iint_{(x,y) \in A_m} dx dy$$

We now find the area of $A_m$. The area where $(\sqrt{x} - \sqrt{y})^2 \leq m$ is the same as the area where $\sqrt{x} - \sqrt{y} \leq \sqrt{m}$, as $x \geq y$. Notice that $\sqrt{x} - \sqrt{y} \leq \sqrt{m}$ implies $x \leq (\sqrt{m} + \sqrt{y})^2$ and we have $x - y = m + 2\sqrt{my}$. Let $\hat{y} = x + y$ and $\hat{x} = x - y$. Then we have $y = \frac{1}{2}(\hat{y} - \hat{x})$ which gives us $\hat{x} - m \leq \sqrt{2m(\hat{y} - \hat{x})}$. So either $\hat{x} \leq m$ or $(\hat{x} - m)^2 \leq 2m(\hat{y} - \hat{x})$. For the latter case, we have $\hat{y} \geq \frac{1}{2m}\hat{x}^2 + \frac{1}{2}m$. We can express the constraints of $A$ as $\hat{y} \leq 1$ and $0 \leq \hat{y} - \hat{x} \leq \hat{y} + \hat{x} \leq 2$, the latter can be decomposed as $\hat{y} \geq \hat{x} \geq 0$. When $\hat{x} \leq m$, all points $(\hat{x}, \hat{y})$ where $0 \leq \hat{x} \leq \hat{y} \leq 1$ are sufficient, and for $\hat{x} \geq m$, we see that $\frac{1}{2m}\hat{x}^2 + \frac{1}{2}m \geq \hat{x}$ so all points $\frac{1}{2m}\hat{x}^2 + \frac{1}{2}m \leq \hat{y} \leq 1$ are sufficient.

Expressing this as an integral, we have

$$\frac{1}{2}F(m) = \iint_{(\hat{x},\hat{y}) \in \Phi(A_m)} d\hat{x} d\hat{y} = \int_0^m (1 - \hat{x}) d\hat{x} + \int_m^1 \max\left(0, 1 - \frac{1}{2m}\hat{x}^2 - \frac{1}{2}m\right) d\hat{x}$$

$$= m\left(1 - \frac{m}{2}\right) + \int_m^{\sqrt{2m-m^2}} \left(1 - \frac{1}{2m}\hat{x}^2 - \frac{1}{2}m\right) d\hat{x}$$

$$= \sqrt{2m - m^2}\left(1 - \frac{m}{2}\right) - \frac{1}{2m}\int_m^{\sqrt{2m-m^2}} \hat{x}^2 d\hat{x}$$

$$= \frac{1}{2m}\sqrt{2m - m^2}^3 - \frac{1}{6m}\left(\sqrt{2m - m^2}^3 - m^3\right)$$

$$= \frac{1}{3m}\sqrt{2m - m^2}^3 + \frac{m^2}{6} = \frac{1}{3}\sqrt{m}\sqrt{2 - m}^3 + \frac{m^2}{6}$$

where $\Phi$ is the transformation $(x, y) \to (\frac{1}{2}(x - y), \frac{1}{2}(x + y))$ and we remove a factor of 2 from the Jacobian in our transformation. The second equality uses $1 - \frac{1}{2m}\hat{x}^2 - \frac{1}{2}m \leq 0$ when $\hat{x} \geq \sqrt{2m - m^2}$. Taking the derivative, we have

$$p(m) = \frac{d}{dm}F(m) = \frac{1}{3\sqrt{m}}\sqrt{2 - m}^3 + 2\sqrt{m}\sqrt{2 - m}^2 \cdot \left(-\frac{1}{2\sqrt{2 - m}}\right) + \frac{2m}{3}$$

$$= \frac{1}{3\sqrt{m}}\sqrt{2 - m}^3 - \sqrt{m}\sqrt{2 - m} + \frac{2m}{3}$$

$\square$

The uniform distribution assumption is clearly not realistic, but we claim that power scaling is natural. Consider another class of datasets with distribution of $(p_1, p_2)$ weighted by $(p_1 + p_2)^n$ for $n > 0$. This arbitrarily downweights questions where both $p_1$ and $p_2$ are low, which are questions where the model has low confidence and considers several responses. We again observe that $p(m) \propto m^{-1/2}$ when $m \to 0$.

**Lemma 7.** *Let $g(p_1, p_2) \propto (p_1 + p_2)^n$ for $n > 0$, then*

$$p(m) \propto -(n+2)m^{n+1} + \frac{1-m}{\sqrt{2-m}\sqrt{m}} + 2^{-(n+1)} \sum_{i=0}^{n+1} \binom{n+1}{i} \frac{n+2}{2i+1} m^{n+1}$$

$$- 2^{-(n+1)} \sum_{i=0}^{n+1} \binom{n+1}{i} \frac{1}{2i+1} \frac{d}{dm} \sqrt{2-m}^{2i+1} m^{n-i+3/2}$$

*which implies that $\lim_{m \to 0^+} p(m) \propto \frac{1}{\sqrt{m}}$.*

*Proof.* Suppose we have $(x+y)^n$

$$F(m) = \iint_{(x,y) \in A_m} g(x,y) dx dy = \iint_{(x,y) \in A_m} \frac{1}{Z}(x+y)^n dx dy$$

for normalizing constant $Z$, and from Theorem 6, we have

$$F(m) \propto \int_0^m \int_m^1 \hat{y}^n d\hat{y} d\hat{x} + \int_m^{\sqrt{2m-m^2}} \int_{\frac{1}{2m}\hat{x}^2 + \frac{1}{2}m}^1 \hat{y}^n d\hat{y} d\hat{x}$$

Equivalently, we have

$$F(m) \propto \int_m^{\sqrt{2m-m^2}} \left( 1 - \left( \frac{1}{2m}\hat{x}^2 + \frac{1}{2}m \right)^{n+1} \right) d\hat{x} + \int_0^m 1 - m^{n+1} d\hat{x}$$

$$= \sqrt{2m - m^2} - (2m)^{-(n+1)} \sum_{i=0}^{n+1} \binom{n+1}{i} m^{2(n+1-i)} \int_m^{\sqrt{2m-m^2}} \hat{x}^{2i} d\hat{x} - m^{n+2}$$

$$= \sqrt{2m - m^2} - 2^{-(n+1)} m^{-(n+1)} \sum_{i=0}^{n+1} \binom{n+1}{i} \frac{m^{2(n+1-i)}}{2i+1} \left( \sqrt{2m-m^2}^{2i+1} - m^{2i+1} \right) - m^{n+2}$$

$$= \sqrt{2m - m^2} - 2^{-(n+1)} \sum_{i=0}^{n+1} \binom{n+1}{i} \frac{1}{2i+1} \left( \sqrt{2-m}^{2i+1} m^{n-i+3/2} - m^{n+2} \right) - m^{n+2}$$

Taking the derivative, we have

$$p(m) = \frac{d}{dm} F(m) \propto -(n+2)m^{n+1} + \frac{1-m}{\sqrt{2-m}\sqrt{m}} + 2^{-(n+1)} \sum_{i=0}^{n+1} \binom{n+1}{i} \frac{n+2}{2i+1} m^{n+1}$$

$$- 2^{-(n+1)} \sum_{i=0}^{n+1} \binom{n+1}{i} \frac{1}{2i+1} \frac{d}{dm} \sqrt{2-m}^{2i+1} m^{n-i+3/2}$$

We have $\frac{1}{\sqrt{m}}$ scaling (up to constant) when $m \to 0$. The first and third term decay to 0 as $m \to 0$ from the $m^{n+1}$ term. For the last term

$$\frac{d}{dm} \sqrt{2-m}^{2i+1} m^{n-i+3/2} = (n - i + \frac{3}{2})\sqrt{2-m}^{2i+1} m^{n-i+1/2} - \frac{2i+1}{2\sqrt{2-m}} \sqrt{2-m}^{2i} m^{n-i+3/2}$$

As $2 - m$ approaches 2 as $m \to 0$ and $i$ is at most $n + 1$, the first term is at best on the order of $m^{-1/2}$ (again giving us $\frac{1}{\sqrt{m}}$ scaling), while the second term is at best on the order of $m^{1/2}$ which decays to 0.[7]   □

Suppose we could observe an even faster convergence. Since questions with low margin converge the slowest, such a dataset should heavily downweight low-margin questions, so we consider $g(p_1, p_2) = (\sqrt{p_1} - \sqrt{p_2})^{2n}$ for $n > 0$. We can attempt to arbitrarily down-weight $p_1 - p_2$ by increasing $n$. However, we still have power law scaling, giving us the desired predictable gains.

---

[7] We get the full $m^{-1/2}$ term is $\left( \frac{1-m}{\sqrt{2-m}} - \frac{2^{-(n+2)}}{2n+3} \sqrt{2-m}^{2n+3} \right) m^{-1/2}$

**Lemma 8.** *Let $g(p_1, p_2) \propto (\sqrt{p_1} - \sqrt{p_2})^{2n}$ for $n > 0$, then*

$$p(m) \propto -m^{n+1} \int_0^{\pi/2} (1 + \cos\theta)^n \cos\theta d\theta + \frac{m^{n-1}}{3} \left( m^2 + (1 - 2m)\sqrt{2m - m^2} \right)$$

*which implies that $\lim_{m \to 0^+} p(m) \propto m^{n-1/2}$.*

*Proof.* Suppose we have $g(x, y) = (\sqrt{x} - \sqrt{y})^{2n}$

$$F(m) = \iint_{(x,y) \in A_m} g(x, y) dx dy = \iint_{(x,y) \in A_m} \frac{1}{Z}(x + y - 2\sqrt{xy})^n dx dy$$

for some normalizing factor $Z$ and from Theorem 6, we have

$$F(m) \propto \underbrace{\int_0^m \int_m^1 \left( \hat{y} - \sqrt{\hat{y}^2 - \hat{x}^2} \right)^n d\hat{y} d\hat{x}}_{I_1(m)} + \underbrace{\int_m^{\sqrt{2m-m^2}} \int_{\frac{1}{2m}\hat{x}^2 + \frac{1}{2}m}^1 \left( \hat{y} - \sqrt{\hat{y}^2 - \hat{x}^2} \right)^n d\hat{y} d\hat{x}}_{I_2(m)}$$

We have $p(m) = \frac{d}{dm} F(m) \propto \frac{d}{dm} I_1(m) + \frac{d}{dm} I_2(m)$. Using the Leibniz Integral rule, we have

$$\frac{d}{dm} I_1(m) = \int_m^1 \left( \hat{y} - \sqrt{\hat{y}^2 - m^2} \right)^n d\hat{y} + \int_0^m \frac{\partial}{\partial m} \int_m^1 \left( \hat{y} - \sqrt{\hat{y}^2 - x^2} \right)^n d\hat{y} d\hat{x}$$

$$= \int_m^1 \left( \hat{y} - \sqrt{\hat{y}^2 - m^2} \right)^n d\hat{y} - \int_0^m \left( m - \sqrt{m^2 - x^2} \right)^n d\hat{x}$$

and

$$\frac{d}{dm} I_2(m) = \frac{1 - m}{\sqrt{2m - m^2}} \int_1^1 \left( \hat{y} - \sqrt{\hat{y}^2 - 2m + m^2} \right)^n d\hat{y} - \int_m^1 \left( \hat{y} - \sqrt{\hat{y}^2 - m^2} \right)^n d\hat{y}$$

$$+ \int_m^{\sqrt{2m-m^2}} \frac{\partial}{\partial m} \int_{\frac{1}{2m}\hat{x}^2 + \frac{1}{2}m}^1 \left( \hat{y} - \sqrt{\hat{y}^2 - \hat{x}^2} \right)^n d\hat{y} d\hat{x}$$

$$= -\int_m^1 \left( \hat{y} - \sqrt{\hat{y}^2 - m^2} \right)^n d\hat{y} + \int_m^{\sqrt{2m-m^2}} \frac{1}{2} \left( \frac{\hat{x}^2}{m^2} - 1 \right) \left( \frac{\hat{x}^2}{2m} + \frac{m}{2} - \sqrt{\left( \frac{\hat{x}^2}{2m} + \frac{m}{2} \right)^2 - \hat{x}^2} \right)^n d\hat{x}$$

$$= -\int_m^1 \left( \hat{y} - \sqrt{\hat{y}^2 - m^2} \right)^n d\hat{y} + \frac{m^n}{2} \int_m^{\sqrt{2m-m^2}} \left( \frac{\hat{x}^2}{m^2} - 1 \right) d\hat{x}$$

We then have

$$p(m) \propto -\int_0^m \left( m - \sqrt{m^2 - x^2} \right)^n d\hat{x} + \frac{m^n}{2} \int_m^{\sqrt{2m-m^2}} \left( \frac{\hat{x}^2}{m^2} - 1 \right) d\hat{x}$$

$$= -m \int_0^{\pi/2} \left( m - \sqrt{m^2 - m^2 \sin^2\theta} \right)^n \cos\theta d\theta + \frac{m^n}{2} \left( \frac{2}{3}m - \sqrt{2m - m^2} + \frac{\sqrt{2m - m^2}^3}{3m^2} \right)$$

$$= -m^{n+1} \int_0^{\pi/2} (1 + \cos\theta)^n \cos\theta d\theta + \frac{m^{n-1}}{3} \left( m^2 + (1 - 2m)\sqrt{2m - m^2} \right)$$

For the first term, we define $\theta$ via $\hat{x} = m \sin\theta$ As $m \to 0$, we see that the very last term dominates and scales as $m^{n-1/2}$. Functions proportional to $m^{n-1/2}$ have Laplace Transforms that are also power laws with rate $m^{-n-1/2}$.

$\square$

## C.3. Proof of Proposition 3

The fixed allocation objective is

$$\min \int_0^1 \exp(-mx_m)p(m)dm$$

under the constraint that $\int_0^1 x_m dm = \bar{x}$ and $x_m \in \mathbb{N}$. We make the simplifying assumption that $x(m) \in [0, \infty)$ to avoid integer programming. As our error models, $\exp(-mx_m)$, are smooth and monotone, this continuous relaxation only changes results by a negligible rounding error.

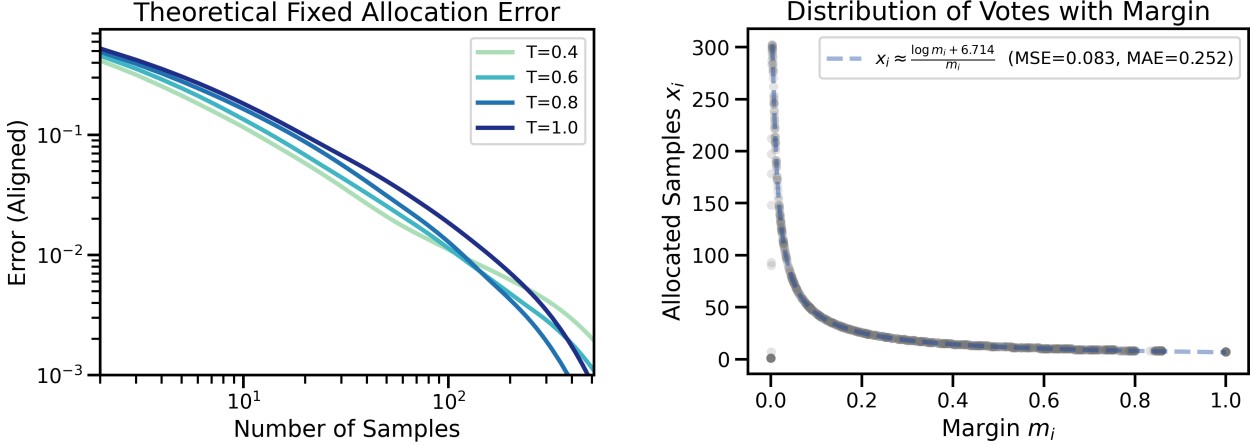

*Figure 10.* We use the error model $e^{-m_i x_{m_i}}$ using margins extracted from running Llama-3.2-3B on GSM8K. (Left) We observe weak power-law scaling initially that tapers off in the high-sample regime. (Right) Samples closely follow the theoretical distribution from Lagrangian optimality.

Under Lagrangian optimality, the solution is of the following form:

$$x_m = \begin{cases} m^{-1}(\log m - \log \lambda) & \text{if } m \geq \lambda \\ 0 & \text{if } m < \lambda \end{cases}$$

for some $\lambda > 0$. The budget constraint then becomes

$$\bar{x} = \int_\lambda^1 (\log m - \log \lambda) \frac{p(m)}{m} dm$$

and the error becomes

$$\text{err}(\bar{x}, \mathcal{D}) = \int_0^\lambda p(m)dm + \lambda \int_\lambda^1 \frac{p(m)}{m} dm$$

Suppose the distribution of margin is of the form $p(m) = (1 - \alpha) \cdot m^{-\alpha}$ for some $\alpha \in (0, 1)$. Then the budget constraint is

$$\bar{x} = (1 - \alpha) \int_\lambda^1 m^{-1-\alpha} \log m \, dm - (1 - \alpha) \log \lambda \int_\lambda^1 m^{-1-\alpha} dm$$

$$= \frac{1 - \alpha}{\alpha} \left[ m^{-\alpha} \log m \right]_1^\lambda + \frac{1 - \alpha}{\alpha} \int_\lambda^1 m^{-1-\alpha} dm - (1 - \alpha) \log \lambda \int_\lambda^1 m^{-1-\alpha} dm$$

$$= \frac{1 - \alpha}{\alpha^2} \lambda^{-\alpha} - \frac{1 - \alpha}{\alpha^2} + \frac{1 - \alpha}{\alpha} \log \lambda$$

As $\bar{x} \to \infty$, we see that $\lambda$ scales proportional to $\bar{x}^{-\frac{1}{\alpha}}$. We can also express the error for

$$\text{err}(\bar{x}, \mathcal{D}) = (1 - \alpha) \int_0^\lambda m^{-\alpha} dm + (1 - \alpha)\lambda \int_\lambda^1 m^{-1-\alpha} dm$$

$$= \frac{1}{\alpha} \lambda^{1-\alpha} - \frac{1 - \alpha}{\alpha} \lambda$$

Again, as $\bar{x} \to \infty$, we have that our error scales as $\lambda^{1-\alpha}$ and therefore $\bar{x}^{-\frac{1-\alpha}{\alpha}}$. In the special case where $\alpha = \frac{1}{2}$, we have that error scales as $\bar{x}^{-1}$ as $\bar{x} \to 0$.

From Proposition 3, questions with very small margins, less than $\lambda$, are ignored since allocating to them is inefficient. Figure 10 confirms the predicted allocation and convergence across synthetic datasets, and we observe strong accelerated power-law scaling behavior on benchmarks.

### C.4. Proof of Corollary 4

Anand Jain et al. (2022) showed that PPR-1v1 is asymptotically optimal per question. We briefly show that this extends to the dataset setting. First, from Shah et al. (2020), we have

> **Theorem 9.** *For $\delta \in (0, 1)$ and stopping condition $\mathcal{S}_\delta$, the expected number of samples is at least*
>
> $$x(\mathcal{S}_\delta, q_i) \geq \sup_{\rho:\arg\max(\rho) \neq \arg\max \mu(\cdot|q)} \frac{1}{\mathsf{KL}(\mu(\cdot \mid q), \rho)} \ln\left(\frac{1}{2.4\delta}\right) := \mathsf{LB}(\delta, q_i)$$
>
> *where $\rho$ is a categorical distribution with the same support as $\mu(\cdot \mid q)$.*

and from Anand Jain et al. (2022), we have

> **Theorem 10.** *Let the stopping criteria of PPR-1v1 be $\mathcal{S}_\delta^P$ for any $\delta$. Then $\lim_{\delta \to 0^+} \frac{x(\mathcal{S}_\delta^P, q_i)}{\mathsf{LB}(\delta, q_i)} = 1$.*

First, at best, we can have exponential error scaling as any specific question has exponential error scaling. Let $q_i = \arg\min_{q \in \mathcal{D}} \sup_{\rho:\arg\max(\rho) \neq \arg\max \mu(\cdot|q)} \frac{1}{\mathsf{KL}(\mu(\cdot \mid q), \rho)}$ and $\rho_i$ be the corresponding distribution. PPR-1v1 achieves the asymptotically optimal exponential scaling as

$$\lim_{\delta \to 0^+} \frac{\mathsf{KL}(\mu(\cdot \mid q_i), \rho_i)}{\ln(\frac{1}{2.4\delta})} \mathbb{E}_{q_j \sim \mathsf{Unif}(\mathcal{D})} \left[x(\mathcal{S}_\delta^P, q_j)\right] \leq \lim_{\delta \to 0^+} \mathbb{E}_{q_j \sim \mathsf{Unif}(\mathcal{D})} \left[\frac{x(\mathcal{S}_\delta^P, q_j)}{\mathsf{LB}(\delta, q_j)}\right] = 1.$$

# D. Additional results on margin correlation

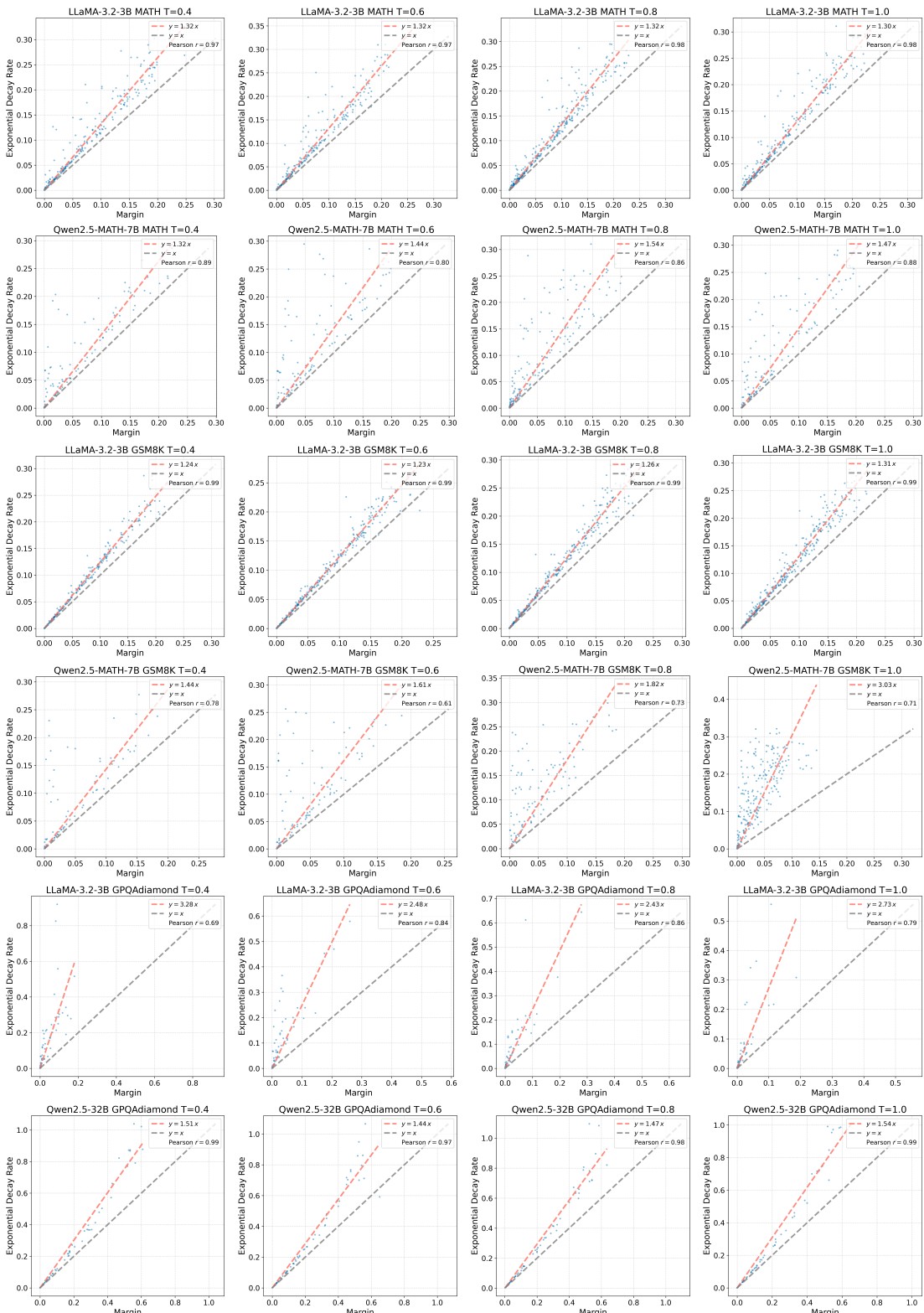

*Figure 11.* We compare the empirical decay rate with margin, and observe high Pearson correlation. This substantiates our theoretical error model in Section 3

# E. Additional results on dataset scaling laws

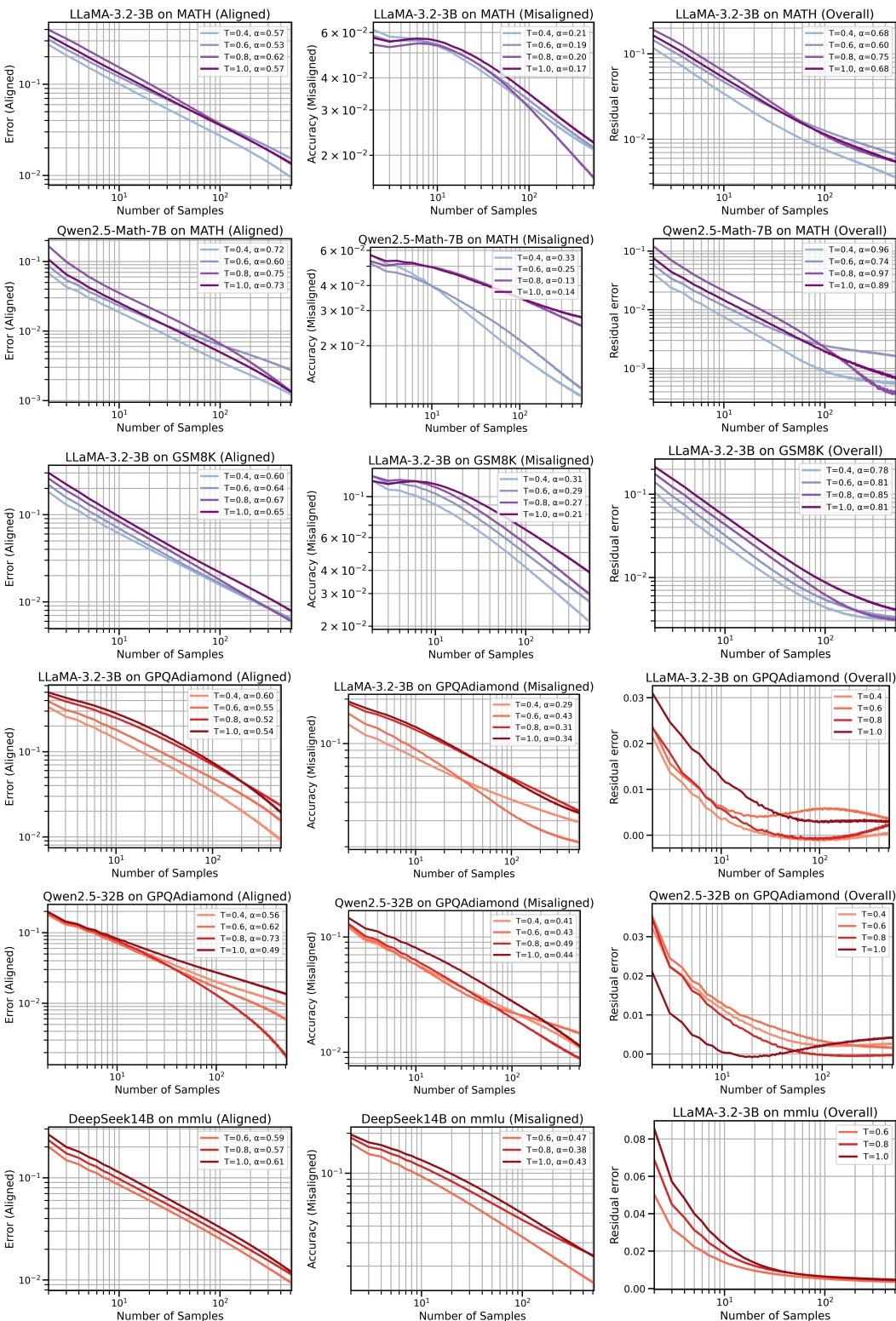

*Figure 12.* We observe strong power law scaling on aligned questions and weaker power law scaling on misaligned questions. For free-response benchmarks (blue), power-law scaling is prominent up to 100 samples. For multiple-choice benchmarks (red), we cannot reliably predict performance.

# F. Additional results on `Blend-ASC`

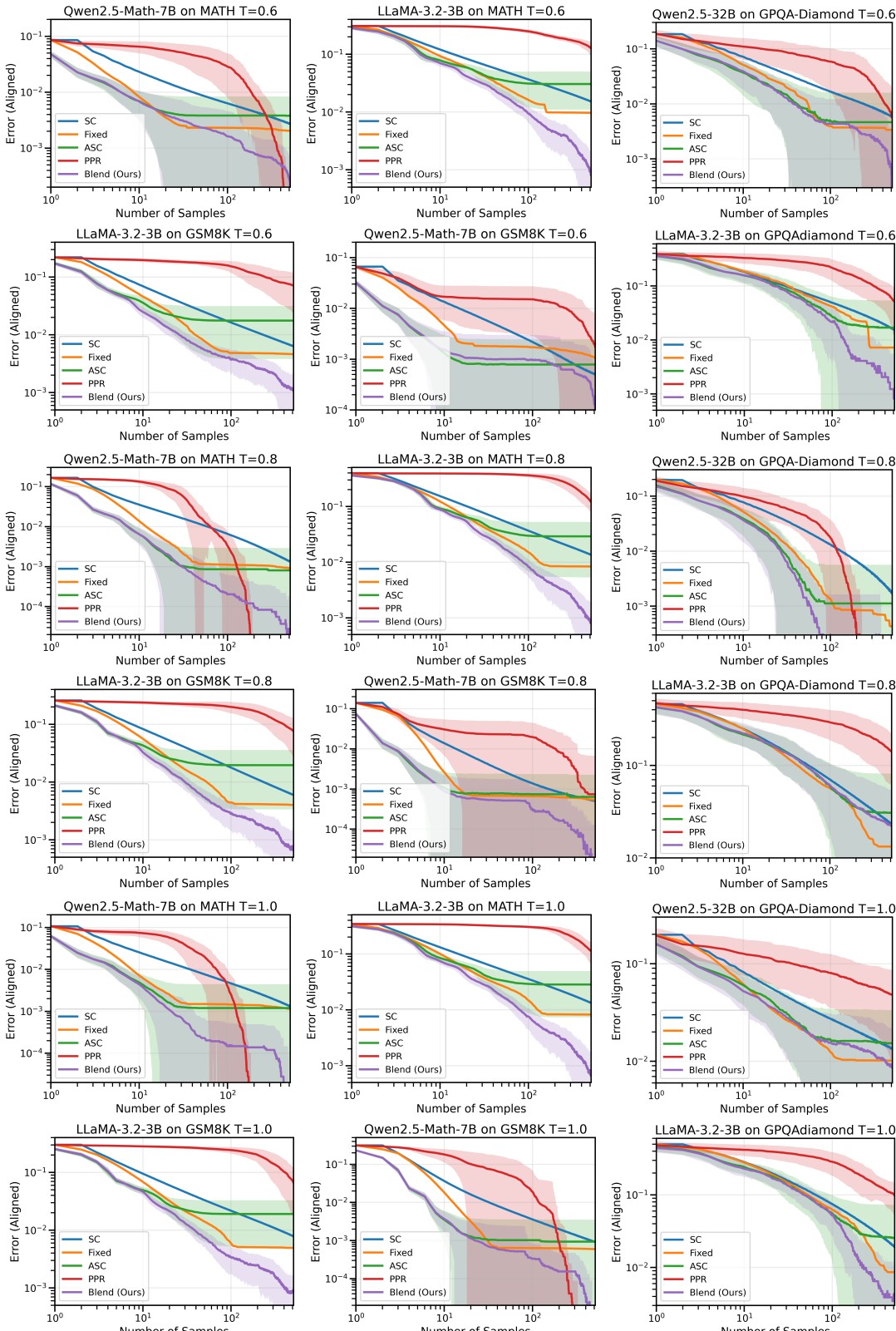

*Figure 13.* We compare all Self-Consistency variants on mode estimation across several temperatures.

| SC@n | Algorithm | GSM8K | | MATH | | GPQA-Diamond | | Average Improvement |
|---|---|---|---|---|---|---|---|---|
| | | Llama-3B | Qwen-Math | Llama-3B | Qwen-Math | Llama-3B | Qwen-32B | |
| 64 | Fixed-Allocation | 23 | 13 | 28 | 11 | 56 | 30 | 2.39 × |
| | Adaptive SC | 24 | 7 | 31 | 10 | 45 | 21 | 2.78× |
| | **Blend-ASC (Ours)** | 13 | 6 | 19 | 9 | 38 | 18 | **3.73×** |
| 128 | Fixed-Allocation | 32 | 75 | 43 | 14 | 81 | 42 | 2.68× |
| | Adaptive SC | 128 | 10 | 77 | 14 | 67 | 33 | 2.33× |
| | **Blend-ASC (Ours)** | 19 | 9 | 30 | 11 | 57 | 25 | **5.09×** |

*(a)* Sample efficiency at temperature 0.6.

| SC@n | Algorithm | GSM8K | | MATH | | GPQA-Diamond | | Average Improvement |
|---|---|---|---|---|---|---|---|---|
| | | Llama-3B | Qwen-Math | Llama-3B | Qwen-Math | Llama-3B | Qwen-32B | |
| 64 | Fixed-Allocation | 15 | 9 | 18 | 10 | 27 | 15 | 4.09× |
| | Adaptive SC | 13 | 6 | 16 | 7 | 33 | 13 | 4.36× |
| | **Blend-ASC (Ours)** | 11 | 6 | 14 | 7 | 26 | 8 | **5.33×** |
| 128 | Fixed-Allocation | 34 | 14 | 45 | 19 | 77 | 37 | 3.40× |
| | Adaptive SC | 128 | 11 | 77 | 13 | 102 | 29 | 2.13× |
| | **Blend-ASC (Ours)** | 22 | 9 | 31 | 12 | 80 | 25 | **4.29×** |

*(b)* Sample efficiency at temperature 0.8.

| SC@n | Algorithm | GSM8K | | MATH | | GPQA-Diamond | | Average Improvement |
|---|---|---|---|---|---|---|---|---|
| | | Llama-3B | Qwen-Math | Llama-3B | Qwen-Math | Llama-3B | Qwen-32B | |
| 64 | Fixed-Allocation | 23 | 17 | 25 | 11 | 53 | 23 | 2.53× |
| | Adaptive SC | 21 | 10 | 31 | 9 | 49 | 30 | 2.56× |
| | **Blend-ASC (Ours)** | 16 | 9 | 20 | 8 | 49 | 21 | **3.12×** |
| 128 | Fixed-Allocation | 36 | 21 | 41 | 15 | 82 | 37 | 3.31× |
| | Adaptive SC | 128 | 13 | 68 | 13 | 87 | 43 | 2.18× |
| | **Blend-ASC (Ours)** | 25 | 12 | 29 | 10 | 90 | 26 | **4.00×** |

*(c)* Sample efficiency at temperature 1.0.

*Table 2.* Comparison of sample efficiency across adaptive methods under different temperatures.

