# OpenReview forum: "Optimal Self-Consistency for Efficient Reasoning with Large Language Models"
_ICML.cc/2026/Conference — ICML 2026 regular_

### Official Review · Reviewer_WUJP · 2026-03-02

**Soundness:** 3
**Presentation:** 2
**Significance:** 3
**Originality:** 3
**Overall Recommendation:** 4
**Confidence:** 3

**Summary:**

In this paper, the author privide a comprehensive analysis of SC's scaling behavior and its variants by framing SC as an empirical mode estimation and majority vote problem which shows a power-law scaling behavior for SC across datasets. Building on these theoretical insights, the authors propose Blend-ASC. Blend-ASC achieves state-of-the-art sample efficiency, requiring 4.8 times fewer samples than vanilla SC on average. Additionally, the method is highly practical as it is hyperparameter-free, supports batching, and can flexibly fit any given sample budget.

**Compliance With Llm Reviewing Policy:**

Affirmed.

**Final Justification:**

I think the authors have addressed my concerns and I decide to maintain my positive score.

**Key Questions For Authors:**

1. The axes-labels in the left and middle panels are incorrectly labeled, showing impossible values such as margin > 1 and samples < 1, should the x-axis be "margin" and the y-axis be "density"?

**Limitations:**

See weakness

**Strengths And Weaknesses:**

Strength

1.This paper formalizes Self-Consistency (SC) as empirical mode estimation. For per-question scaling, the authors derive a tighter exponential error decay bound based on a square-root-based margin. Extending this to dataset-level performance, they apply the Laplace transform to prove that the margin distribution naturally leads to the power-law scaling of errors, which is primarily driven by low-margin questions.

Weakness

1.The method design lacks novelty：Blend-ASC is essentially a linear interpolation of two existing methods, ASC and PPR-1v1.

2.Presentation and Technical Typos:

    a. Figure 3: The X-axes in the left and middle panels are incorrectly labeled, showing impossible values such as margin > 1 and samples < 1.

    b. Line 307: In the cost analysis, should 37.12 * 1.8 ¡ 70 be 37.12 * 1.8 < 70

    c. Figure 4: In the bottom-left sub-figure, only three curves are visible despite multiple experimental settings. Is this a plotting error, or do some curves perfectly overlap?

---

> ### Author Rebuttal · Authors · 2026-03-31
>
> We thank the reviewer WUJP for highlighting our algorithm's **practical design and robust theoretical formalization** that spans per-question and dataset performance. We provide below a point-by-point clarification to address their specific questions.
>
> > **1. The method design lacks novelty：Blend-ASC is essentially a linear interpolation of two existing methods, ASC and PPR-1v1.**
>
> We do not believe our method is just a simple interpolation of two methods. First, PPR-1v1 is a purely statistical mode estimation algorithm that has never been used before in Large Language Models or test-time scaling. This is an innovative insight and connection we discovered through our theoretical framework. Additionally, existing algorithms like ASC lack significant practicality and do not support batching, no hyperparameter tuning, and fixed budgets, while our method includes all. These design choices require a dataset-optimization perspective, which we contribute to the Self-Consistency literature.
>
> > **2. Presentation and Technical Typos. The axes-labels in the left and middle panels are incorrectly labeled, showing impossible values such as margin > 1 and samples < 1, should the x-axis be "margin" and the y-axis be "density"?**
>
> We deeply thank the reviewer for identifying these presentation and technical typos. The reviewer's comments are very helpful in clarifying our communication, and we will edit these. For Figure 3 (left and middle), the axes labels are incorrectly flipped, as you mentioned. For Figure 4, we thank the reviewer for highlighting this small plotting error. The full plot can be seen in the fifth row of Figure 12 in Appendix E, where four lines for T=0.4,0.6,0.8,1.0 can be observed.
>
> We thank the reviewer for the positive feedback and remain at their disposal if further clarifications are needed.

---

> > ### Author Rebuttal · Reviewer_WUJP · 2026-04-02
> >
> > I think the authors have addressed my concerns. I will maintain my positive score.

---

> > > ### Author Response · Authors · 2026-04-03
> > >
> > > We are pleased to read the reviewer’s concerns are fully addressed. We thank again the reviewer for acknowledging that Blend-ASC achieves SOTA sample efficiently and is highly practical. With the novelty of Blend-ASC made clearer, we believe our submission is strengthened and thank the reviewer for helping us improve our work.
> > >
> > > Best regards,
> > >
> > > The authors

---

### Official Review · Reviewer_dsX4 · 2026-03-08

**Soundness:** 2
**Presentation:** 2
**Significance:** 2
**Originality:** 2
**Overall Recommendation:** 4
**Confidence:** 3

**Summary:**

This paper provides a theoretical and empirical study of Self-Consistency. It derives per-question error decay bounds governed by a “margin” between the top-1 and top-2 answer probabilities, and connects dataset-level SC scaling to the margin distribution, explaining why power-law behavior can emerge. Building on these insights, the paper studies sample-allocation strategies (fixed and dynamic) and proposes Blend-ASC, a hyperparameter-free adaptive variant that blends ASC-style confidence ranking with a theoretically grounded PPR-1v1 ranking under a fixed total budget. Experiments show substantially improved sample efficiency, reporting about 4.8× fewer samples than vanilla SC on average at comparable accuracy.

**Compliance With Llm Reviewing Policy:**

Affirmed.

**Final Justification:**

After re-evaluating the paper in light of the authors’ response, I believe my main concerns have been adequately addressed, and I have updated my score accordingly. The rebuttal improves the clarity and support for the main claims.

**Key Questions For Authors:**

* How sensitive are the theory and Blend-ASC to the estimation of (\mu(\cdot\mid q)), top-(k) truncation, and long-tail “unique answer” handling?
* Does the multinomial resampling simulation match on-the-fly adaptive sampling in real decoding?

**Limitations:**

yes

**Strengths And Weaknesses:**

Strengths

* **Solid theoretical framing:** Treating SC as empirical mode estimation makes the analysis principled, and the top-1/top-2 “margin” gives an interpretable handle on per-question scaling.
* **Links per-item and dataset scaling:** Connecting average error to the margin distribution provides a coherent explanation for observed power-law-like SC curves.

Weaknesses

* **Novelty is partly incremental:** Top-1 vs top-2 separability/margin ideas and early-stopping/adaptive SC are well explored; the paper should more crisply separate what is genuinely new (e.g., specific bounds, allocation optimality) from established concepts.
* **Assumptions and approximation gaps:** Several steps rely on approximating the answer distribution (finite samples, top-(k) truncation/merging of rare answers); the impact of these approximations on the theoretical conclusions and on real API usage needs clearer discussion.
* **Simulation-based evaluation limits realism:** Using a large pre-sampled pool to estimate (\mu(\cdot\mid q)) and then multinomial resampling is efficient, but may not capture dependencies induced by prompting, decoding settings, or adaptive sampling on-the-fly.
* **Cost metrics could be broader:** “Samples saved” is informative, but reporting latency / token cost and variance across runs would better reflect real deployment tradeoffs.

---

> ### Author Rebuttal · Authors · 2026-03-31
>
> We thank the reviewer dsX4 for highlighting that **our strong theoretical framing and dataset scaling results.**. We provide below a point-by-point clarification to address their specific questions.
>
> > **1. Novelty is partly incremental: Top-1 vs top-2 separability/margin ideas and early-stopping/adaptive SC are well explored; the paper should more crisply separate what is genuinely new (e.g., specific bounds, allocation optimality) from established concepts.**
>
> We thank the reviewer for proposing to improve the clarity in describing the novelty of our theoretical results. However, we believe that our novelty is not incremental. While adaptive SC methods exist, we provide the first holistic exploration across SC variants grounded in theory.
>
> We highlight that the bulk of our theoretical explorations are all novel. While margin has been identified by other papers, we present the first work that explores margin-based dataset scaling laws, reflecting numerous real-world use cases. Adaptive Self-Consistency methods lack convergence guarantees, and to provide this theory, we identified PPR-1v1, a statistical algorithm never used in LLMs before, and applied it to our setting for optimal convergence rates. We also explore new variants and theories through Fixed-Allocation SC, and show that these methods are inoptimal.
>
> Additionally, while algorithms like ASC explore early-stopping, they lack significant practicality, and do not support batching, no hyperparameter tuning, and fixed budgets, while our method includes all. These design choices require novel dataset-optimization perspectives, which we contribute to the Self-Consistency literature.
>
>
> > **2. Assumptions and approximation gaps: Several steps rely on approximating the answer distribution (finite samples, top-(k) truncation/merging of rare answers); the impact of these approximations on the theoretical conclusions and on real API usage needs clearer discussion.**
>
> Thank you for bringing up these assumptions. We believe they do not affect the theory and have minimal impact on real API usage. Regarding the theory, we use top-k truncation where k can be any large enough integer. So for every question, some k satisfying $\bar p_k<p_2$ exists. We observe that $k$ does not show up in Theorem 1 inequality, except in the $\epsilon$ term, which decays to $0$. The purpose of $k$ is so that we can use a finite "voter" scheme for a vacuously infinite answer LLM. Please see Appendix C.1 to observe how truncating does not affect our theoretical bound besides $\epsilon$.
>
> In practice, we also see that the margin bound in Theorem 1, which assumes top-k truncation, correlates well with the empirical decay. Regarding real API usage, top-k truncation is not used at all.
>
> Finite samples do not affect the theory, as we use the true LLM probability distribution. Under the next question, we discuss the minimal impacts of finite sample simulations in approximating API usage.
>
> > **3. Simulation-based evaluation limits realism: Using a large pre-sampled pool to estimate (\mu(\cdot\mid q)) and then multinomial resampling is efficient, but may not capture dependencies induced by prompting, decoding settings, or adaptive sampling on-the-fly.**
>
> First, we preface that our simulations are close approximations of $\mu(\cdot\mid q)$ using 100 samples, and they have minimal impact on realism. Given the high sample sizes, we are unlikely to miss any significant responses or distribution tail. We are also adding online LLM-based experiments for model/API-based evaluations.
>
> We do not believe prompting and decoding strategies (such as temperature or top_p) sampling have any effect on realism. We note that in typical Self-Consistency applications, prompts and decoding parameters are fixed before generation. These choices affect our token logits and therefore our answer distribution, so if they change, so does $\mu(\cdot\mid q)$. Instead of generating with these settings in an online fashion, we batch generate them beforehand. In fact, the only difference is that with real-time generation, we sample from $\mu(\cdot\mid q)$, while for simulations, we sample from $\hat\mu(\cdot\mid q)$.
>
> We would appreciate more information on the adaptive sampling on-the-fly technique the reviewer mentioned.
>
> > **4. Cost metrics could be broader: “Samples saved” is informative, but reporting latency/token cost and variance across runs would better reflect real deployment tradeoffs.**
>
> Following part 3, we are evaluating token costs using our currently-running LLM-based experiments. We note that we expect token costs to be low as we use few samples (output tokens) and batching can reduce input tokens. Finally, we have shown batched experiments with minimal degradation in Figure 8, which would lead to high throughput.
>
> We thank the reviewer for the constructive feedback and remain at their disposal if further clarifications are needed.

---

> > ### Author Rebuttal · Reviewer_dsX4 · 2026-04-02
> >
> > Thanks for the comprehensive response. My concerns have been adequately addressed and I've updated my score accordingly.

---

> > > ### Author Response · Authors · 2026-04-02
> > >
> > > We are pleased to read the reviewer’s concerns are fully addressed. We thank the reviewer for their constructive feedback and their increased score!
> > >
> > > Best,
> > >
> > > The authors

---

### Official Review · Reviewer_td2E · 2026-03-09

**Soundness:** 3
**Presentation:** 3
**Significance:** 3
**Originality:** 3
**Overall Recommendation:** 4
**Confidence:** 4

**Summary:**

The authors present Blend-ASC, an adaptive self-consistency approach that requires no training or hyperparameter selection.
Blend-ASC adaptively allocates additional generations to challenging queries that need them by `blending' a low-sample (adaptive SC) and asymptotically optimal (prior-posterior ratio) approache to improve performance across regimes.
This blending approach achieves a roughly 4-5x improvement in sample efficient over baseline SC, and ~20% improvement over existing adaptive approaches, when evaluated across a LLaMA and two Qwen models on mathematical reasoning tasks (GSM8K, MATH, and GPQA).
The authors motivate this approach via a power-law scaling analysis grounded in voting theory.

**Compliance With Llm Reviewing Policy:**

Affirmed.

**Final Justification:**

The authors rebuttal has addressed my concerns; I maintain my positive score.

**Key Questions For Authors:**

See weaknesses.

1. What exactly is the protocol for estimating the empirical "true LLM distribution?"

**Limitations:**

Two minor methodological limitations I see are (1) the empirical LLM distribution reliance, and (2) the majority voting assumption. For (1), as we discuss in our weakness section, performance is inherently coupled to the accuracy of this estimation. For (2), while it is sensible to focus on the majority voting formulation of SC, it does limit the generalizability of the findings.

**Strengths And Weaknesses:**

Strengths:
1. Theoretical grounding: the build up through Section 3 and 4 to present the scaling results are compelling, particularly Fig 4 and 7, and the mode estimation / voting theory underpinning are elegant and original.
2. Sample Efficiency: The reported 4-5x improvement over conventional SC is a substantial gain.
3. Hyperparameter-free: Because the method of hyperparameter free and requires no real training / tuning, I believe it is broadly applicable and easily reproducible.

Weaknesses:
1. Use of an empirical "true LLM distribution:" Section 5 briefly describes extracting the "true" LLM distribution to then perform SC via sampling from this distribution. (1) this process is a bit opaque, and would benefit from additional detail and clarity (even if only as an additional subsection to Appendix B), and (2) the motivation for this as a proxy is not very compelling, beyond the cost argument.
2. Lack of long-context evaluation: the evaluated datasets (GSM8K, etc) are all fairly short-context tasks; evaluating this work in the long-context regime is important as many uses of LLMs occur with long-context. Recent work has shown SC exhibits problems with long-context [1], so an accurate accounting of Blend-ASC's limitations is important. That is, if it is not appropriate for long-context, just say that, but if it does work, then that is an important distinction to make!
3. Limited aggregation strategy: While the authors effectively demonstrate Blend-ASC on tasks requiring majority voting, other aggregation strategies exist, such as LLM-as-a-judge (Universal SC [2]) or log-likelihood (Soft-SC [3]).

[1] Byerly, Adam, and Daniel Khashabi. "Self-Consistency Falls Short! The Adverse Effects of Positional Bias on Long-Context Problems." arXiv preprint arXiv:2411.01101 (2025).

[2] Chen, Xinyun, et al. "Universal Self-Consistency for Large Language Models." ICML 2024 Workshop on In-Context Learning.

[3] Wang, Han, et al. "Soft self-consistency improves language models agents." Proceedings of the 62nd Annual Meeting of the Association for Computational Linguistics (Volume 2: Short Papers). 2024.

---

> ### Author Rebuttal · Authors · 2026-03-31
>
> We thank the reviewer td2E for highlighting **the theoretical strength, efficiency gains, and hyperparameter-free design**. We provide below a point-by-point clarification to address their specific questions.
>
> > **1. Use of an empirical "true LLM distribution"**
> >
> To extract the true LLM response, we sample 100 Chain-of-Thought (CoT) responses for each prompt $p$. To simulate an LLM response to $p$, we sample a CoT from this pool and then extract its final answer.
>
> Every LLM has a distribution over final answers. Our simulations are approximations of this distribution using 100 samples per prompt. Given the high sample sizes, the approximation is unlikely to miss any significant modes or distribution tails.
>
> The motivation for this approach is to reduce the cost and time of online generation needed to evaluate models on hundreds of questions with up to a hundred samples each, all run several times for error bars.
>
> > **2. Lack of long-context evaluation**
>
> Thank you for sharing this interesting recent work by Byerly et al. [1] on Self-Consistency in long-context settings. We evaluate the QuALITY benchmark using Qwen-2.5-3B, which Byerly et al. [1] observe to perform worse by -0.3 under majority voting.
>
> We first note that, unlike Self-Consistency, the authors do not use Chain-of-Thought and only output a final multiple choice answer - their prompt explicitly includes "Do not
> provide any explanation." The reason why Self-Consistency is not appropriate is that this leads to a one-token answer, while SC requires reasoning rollouts. In practice, we observe that next-token distributions are sharp, meaning they converge quickly under our theory.
>
> | C@K | T=0.4      | T=0.6      | T=0.8      | T=1.0      |
> | --- | ---------- | ---------- | ---------- | ---------- |
> | 1   | 66.0 ± 0.2 | 66.0 ± 0.2 | 66.0 ± 0.2 | 66.0 ± 0.3 |
> | 15  | 65.9 ± 0.1 | 66.0 ± 0.1 | 66.0 ± 0.2 | 66.0 ± 0.2 |
> | 30  | 65.9 ± 0.1 | 65.9 ± 0.1 | 65.9 ± 0.1 | 65.9 ± 0.1 |
>
> However, CoT distributions are not as sharp as we see below.
>
>
> | Method       | Entropy (bits) | Margin |
> | ------------ | -------------- | ------ |
> | No-CoT T=0.4 | 0.056          | 0.969  |
> | No-CoT T=0.6 | 0.086          | 0.953  |
> | No-CoT T=0.8 | 0.118          | 0.937  |
> | No-CoT T=1.0 | 0.153          | 0.920  |
> | CoT T=0.6    | 1.553          | 0.282  |
> | CoT T=0.8    | 1.636          | 0.256  |
> | CoT T=1.0    | 1.691          | 0.250  |
>
>
> SC and Blend-ASC both improve CoT performance and achieve the best overall performance.
>
> | K   | T=0.6 | T=0.8 | T=1.0 | No-CoT |
> | --- | ----- | ----- | ----- | ------ |
> | 1   | 46.0% | 44.2% | 43.0% | 66.0%  |
> | 5   | 55.3% | 53.6% | 46.5% | 66.0%  |
> | 10  | 60.1% | 59.3% | 59.2% | 66.0%  |
> | 50  | 66.1% | 66.7% | 67.0% | 66.0%  |
> | 75  | 66.8% | 67.3% | 67.5% | 66.0%  |
> | 100 | 67.5% | 67.3% | 67.7% | 66.0%  |
>
> > **3. Limited aggregation strategy**
>
> We emphasize that we focus on algorithmic efficiency and its theory, and not answer extraction. Several open-ended approaches to Self-Consistency could be applied with our method. For example, Jain et al. [2] use semantic clustering to group open-ended prompts. Other adaptations include quantizing or binning encoded prompts in a continuous latent space. General metrics in continuous space can be used to calculate confidence scores for dataset allocation. For Universal Self-Consistency [3], we could use the LLM judge to generate confidence scores based on the answer distribution. We highlight that these approaches would lose our strong theoretical guarantees.
>
> We thank the reviewer for the positive feedback and remain at their disposal if further clarifications are needed.
>
> [1] "Self-Consistency Falls Short! The Adverse Effects of Positional Bias on Long-Context Problems", Byerly and Khahsabi, 2024.
>
> [2] Lightweight Reranking for Language Model Generations, Jain et al., 2024.
>
> [3] Universal Self-Consistency for Large Language Model Generation, Chen et al., 2023.

---

> > ### Author Rebuttal · Reviewer_td2E · 2026-04-03
> >
> > Thank you for the thorough response.
> >
> > Regarding Weakness 1, the clarification is satisfactory, and the cost/time motivation is well-taken. Regarding Weakness 2, I am also withdrawing this concern - the Blend-ASC results are interesting.
> >
> > Regarding Weakness 3, I appreciate the authors' acknowledgment that extensions to other aggregation strategies would come at the cost of the paper's theoretical guarantees. However, I would ask the authors to go further and explicitly acknowledge in the paper that the majority voting assumption is a formal limitation — not all tasks admit a natural majority voting formulation, and this restricts the applicability of Blend-ASC beyond the evaluated benchmarks. A sentence or two in the limitations section to this effect would meaningfully improve the paper's intellectual honesty and help readers understand where the method does and does not apply.

---

> > > ### Author Response · Authors · 2026-04-08
> > >
> > > We thank the reviewer td2E for noting the strong long-context results, which highlight the practical usefulness of Blend-ASC across the evaluated benchmarks.
> > >
> > > We also thank the reviewer for the helpful suggestion and agree that the majority-vote assumption should be stated more explicitly as a formal limitation. To address this, we will add the following text to page 3 after the *"Comparison with prior work"* section:
> > >
> > > "**Majority vote assumption.** Our current framework assumes that candidate answers can be aggregated through a majority-vote-style rule. As a result, Blend-ASC and its theoretical guarantees do not directly apply to tasks where aggregation is unnatural or ill-defined, such as open-ended generation."
> > >
> > > We will also add the following discussion on extensions of our work to the
> > > *"Conclusion and Future Work"* section:
> > >
> > > "A natural direction for future work is to extend Blend-ASC beyond majority-vote-style aggregation. We believe the main challenge is to define suitable confidence measures for alternative aggregation strategies. For instance, one could aggregate open-ended outputs using embedding space clustering [1] and define confidence through density or cohesion-based measures over clusters. Some practical components of our approach, such as global, budget-aware allocation, may transfer more broadly across Self-Consistency variants. Establishing analogous theoretical scaling laws in these settings would require assumptions on per-question correctness scaling behavior."
> > >
> > > We appreciate the reviewer’s suggestion and agree that this clarification will improve the paper’s presentation of scope and limitations.
> > >
> > > [1] Lightweight Reranking for Language Model Generations, Jain et al., 2024.

---

### Official Review · Reviewer_7ifP · 2026-03-11

**Soundness:** 3
**Presentation:** 2
**Significance:** 2
**Originality:** 3
**Overall Recommendation:** 4
**Confidence:** 3

**Summary:**

A mathmatically oriented paper, addressing an important domain of error mitigation with fixed model on a dataset.
The authors focus on assymptotic optimal algorithms for small models and non semantic comparison/binning of the replies.
the core contribution is a theoretical one, "a stopping critrea for sampling replies from a LM" implemented with an algorithm for whole dataset optimal budgeted output

**Compliance With Llm Reviewing Policy:**

Affirmed.

**Key Questions For Authors:**

1. Did you compare the presented method for improving the results with model cascading, can this be added to the paper
2. Do you have a way to generalize your approach to more problematic replies distribution?

**Limitations:**

yes

**Strengths And Weaknesses:**

# Strengths

## 1. Dataset-Wide Optimization
Rather than looking at each question in a vacuum, the algorithm manages the entire dataset at once. This is a specific use-case that is understudied and the authors apporach to whole-dataset-budgeting is insightful.

## 2. Solid Mathematical Foundation
The paper moves away from "guesswork" and ad-hoc methods, grounding its logic in proven voting theory and mode estimation. This provides a reliable, theory-backed way to predict how the model will improve as more samples are added.

## 3. Stability for Both Easy and Hard Prompts
The method successfully controls the variance for different types of prompts. It uses a "blended" strategy that stays efficient on easy, high-margin questions while using more advanced logic to stabilize results on difficult, low-margin questions.

# Weaknesses

## 1. Simplified Answer Distribution
The authors rely heavily on math and multiple-choice benchmarks, which is a very relaxed setup compared to the messy reality of open-ended prompts. They don't solve the core real-world issue: how to group and categorize different-looking but identical answers in a natural language flow.

## 2. Cost and Practicality at Scale
The paper focuses on large sample sizes and "asymptotic" behavior. A more standard industry approach to "hard" questions is to send them to a larger model; comparing this "parallel sampling" against "model cascading" would have made the paper much stronger and more applicable.

---

> ### Author Rebuttal · Authors · 2026-03-31
>
> We thank the reviewer 7ifP for highlighting the **originality of our dataset-wide focus and our theoretical contributions to self-consistency**. We provide below a point-by-point clarification to address their specific questions.
>
> > **1. Simplified Answer Distribution. The authors rely heavily on math and multiple-choice benchmarks, which is a very relaxed setup compared to the messy reality of open-ended prompts. They don't solve the core real-world issue: how to group and categorize different-looking but identical answers in a natural language flow.**
>
> We thank the reviewer for this question. This concern pertains to answer extraction, which is orthogonal to our contribution in algorithmic efficiency. Our method is more dependent on the answer distribution and confidence rather than on how answers are compared.
>
> Several open-ended approaches to Self-Consistency could be applied with our method. For example, Jain et al. [1] use semantic clustering while Universal Self-Consistency [2] uses an LLM as a judge. Other adaptations include quantizing or binning encoded prompts in a continuous latent space. General metrics in continuous space can be used to calculate confidence scores for dataset allocation. We emphasize that our contribution is optimal sampling and allocation given answer distribution confidence, not solving the answer extraction problem itself.
>
>
> > **2. Cost and Practicality at Scale. The paper focuses on large sample sizes and "asymptotic" behavior. A more standard industry approach to "hard" questions is to send them to a larger model; comparing this "parallel sampling" against "model cascading" would have made the paper much stronger and more applicable.**
>
>
> We thank the reviewer for identifying this interesting industry approach. First, we emphasize that while we reserve "asymptotic optimality" for our theoretical analysis, our method achieves strong performances in crucial empirical regimes where our goal is to maximize the performance of a single model. For example, Self-Consistency is used with 64 samples per question in DeepSeek-R1 [3] and both 64 and 256 samples for Qwen2.5-MATH [4]. Blend-ASC could significantly reduce the number of samples required while maintaining performance, making it relevant for performance-critical applications on frontier models.
>
> Regarding cascading, this is a complementary approach to Self-Consistency. Cascading fundamentally improves the LLM generator's performance (by switching to a larger, more capable model). This is analogous to sequential scaling in Snell et al. [5], which also aims to improve the LLM generator's performance. In contrast, Self-Consistency ensures that the LLM generator reaches its peak performance, analogous to parallel scaling. We believe that, similar to Snell et al. [5], cascading and Self-Consistency can be naturally combined for optimal compute scaling.
>
> Finally, we note that cascading is outside the scope for our baselines, as our goal is to improve the sample efficiency of Self-Consistency. If the reviewer could point us to specific cascading methods and implementations, we could plausibly consider incorporating cascading into our work.
>
> We thank the reviewer for the positive feedback and remain at their disposal if further clarifications are needed.
>
> [1] Lightweight Reranking for Language Model Generations, Jain et al., 2024.
>
> [2] Universal Self-Consistency for Large Language Model Generation, Chen et al., 2023.
>
> [3] DeepSeek-R1 Incentivizes Reasoning in LLMs Through Reinforcement Learning, Guo et al., 2025.
>
> [4] Qwen2.5-Math Technical Report: Toward Mathematical Expert Model via Self-Improvement, Yang et al., 2024.
>
> [5] Scaling LLM Test-Time Compute Optimally Can be More Effective than Scaling Parameters for Reasoning, Snell et al., 2025.

---

### Decision · Program_Chairs · 2026-04-30

**Decision:**

Accept (regular)

**Comment:**

This paper presents a theoretical and empirical analysis of Self-Consistency (SC), framing it as a mode estimation problem and deriving margin-based error decay and dataset-level scaling laws. Based on these insights, the authors propose Blend-ASC, a hyperparameter-free adaptive sampling method that allocates generation budget across a dataset, achieving substantial gains in sample efficiency on mathematical reasoning benchmarks.

Reviewers appreciate the strong theoretical grounding, the novel dataset-level optimization perspective, and the practical advantages of the method (e.g., no tuning, efficient allocation). However, they also note limitations, including simplified evaluation settings, assumptions such as majority voting and approximated answer distributions, and limited exploration of long-context and open-ended scenarios. Some concerns about clarity, novelty positioning, and evaluation realism were also raised.

In the rebuttal, the authors addressed these concerns with additional clarifications, experiments, and by explicitly acknowledging key limitations. Reviewers found these responses largely satisfactory and maintained positive assessments. Overall, given the solid technical contributions and effective rebuttal, I recommend acceptance.